# Learning Bayesian Nash Equilibrium in Auction Games via Approximate Best Response

Kexin Huang [1]  Ziqian Chen [2]  Xue Wang [2]  Chongming Gao [1]  Jinyang Gao [2]  Bolin Ding [2]  Xiang Wang [1]

## Abstract

Auction plays a crucial role in many modern trading environments, including online advertising and public resource allocation. As the number of competing bidders increases, learning Bayesian Nash Equilibrium (BNE) in auctions faces significant scalability challenges. Existing methods often experience slow convergence in large-scale auctions. For example, in a classic symmetric auction setting, the convergence rate depends on the number of bidders quadratically. To address this issue, we propose the *Approximate Best Response Gradient* method, a new approach for learning BNE efficiently in auction games. We leverage an analytic solution for gradient estimation to enable efficient gradient computation during optimization. Moreover, we introduce the *Best Response Distance* objective, which serves as an upper bound of approximation quality to BNE. By optimizing the new objective, our method is proven to achieve a local convergence rate independent of bidder numbers and circumvent the traditional quadratic complexity in the classic symmetric setting. Extensive experiments across various auction formats demonstrate that our approach accelerates convergence and enhances learning efficiency in complex auction settings.

## 1. Introduction

Auctions serve as a fundamental mechanism for resource allocation and price discovery. They are extensively applied across diverse industries, such as online advertising for ad impression allocation (Shen et al., 2015) and the distribution of public resources (*e.g.,* spectrum bands (Milgrom, 2000),

[1]University of Science and Technology of China, Hefei, China [2]Independent Researcher. Correspondence to: Chongming Gao <chongminggao@ustc.edu.cn>, Xiang Wang <xiangwang1223@gmail.com>.

*Proceedings of the 42nd International Conference on Machine Learning*, Vancouver, Canada. PMLR 267, 2025. Copyright 2025 by the author(s).

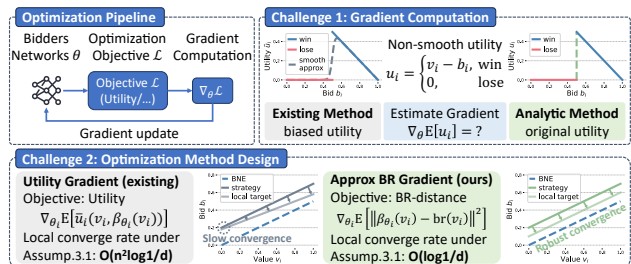

*Figure 1.* This work addresses two key challenges in learning BNE for auction games: (1) gradient computation and (2) optimization method design. We utilize an analytic gradient solution to avoid the biased utility function and propose a new optimization method, approximate BR gradient, for accelerated convergence rate.

mineral rights (Milgrom & Weber, 1982)), to facilitate efficient and transparent market outcomes. At the core of the auction is a competitive bidding process, where bidders submit bids to purchase goods or services from auctioneers, with the goal of maximizing their utility. Wherein, bidders adjust their bidding strategies by estimating values or equilibrium to avoid potential losses (Balseiro et al., 2015), while auctioneers analyze the equilibrium to assess the auction mechanism's effectiveness and predict market behavior (Krishna, 2010), aiding in the evaluation of overall market conditions (Milgrom & Weber, 1982). To formalize this equilibrium behavior, the auction can be modeled as an incomplete-information game (Harsanyi, 1967), where the equilibrium concept is captured by the Bayesian Nash Equilibrium (BNE). However, solving BNE in auction games is computationally hard, classified as PPAD-complete (Filos-Ratsikas et al., 2021; Filos-Ratsikas et al., 2024), with only limited closed-form solutions available in simplified settings (Kaplan & Zamir, 2012).

This fundamental challenge has attracted great interest in integrating machine learning techniques with game theory (Bichler et al., 2021; Huang et al., 2024) to learn the equilibrium bidding strategies. Early efforts have focused on discretized approximations of auction games, using methods such as no-regret learning (Hartline et al., 2015; Kolumbus & Nisan, 2022; Ahunbay & Bichler, 2025), reinforcement learning (Feng et al., 2021; Banchio & Skrzypacz, 2022), and online optimization (Bichler et al., 2023a), to approxi-

mate BNE strategies. However, in discretized auction games, achieving more accurate predictions requires finer discretization, which will significantly increase computational cost (Bichler et al., 2023a). This limits both the precision and scalability of the discrete learning frameworks.

Distinguishing from discretized auctions where bidding strategies map categorical values to bids, continuous auctions model bidding strategies as functions of continuous values. Such differences present two challenges to current learning methods. Specifically, the first challenge lies in the computation of gradients, which is crucial for continuous optimization. Due to the non-smooth nature of utility functions in auction games, gradients cannot be computed directly (Bichler et al., 2021), necessitating advanced gradient estimation techniques, such as smoothing approximations of the utility function (Li & Wellman, 2021; Kohring et al., 2023). However, these approximations will modify the original utility, introducing model bias in the learned strategies (Kohring et al., 2023). This issue is compounded by the challenge of designing optimization methods to effectively utilize the estimated gradients. Specifically, existing methods (Bichler et al., 2021; Kohring et al., 2023) that apply gradient ascent with the utility's gradient suffer from slow convergence as the number of bidders increases.

In this work, we address both the gradient computation and optimization method design challenges in continuous auctions, which is summarized in Fig. 1. We introduce a new gradient estimation method with an analytic solution for the utility's gradients, avoiding the modified utility and model bias in previous approaches. Leveraging this analytic gradient, we identify that the slow convergence of existing methods stems from the inappropriate optimization objective, *i.e.,* the utility function, whose convergence rate degrades quadratically as the number of bidders increases in a classic symmetric auction setting. To address this, we propose a new optimization objective called *Best Response Distance*, which measures the gap between current bidding strategies and their local optimal counterparts, providing an upper bound on the approximation quality to BNE. By optimizing this objective, our proposed *Approximate Best Response Gradient* method achieves a local convergence rate independent of the number of bidders in the symmetric auction setting. Extensive experiments further validate the significant improvement in convergence speed across various auction scenarios.

Our contribution is summarized below:

- We introduce a new gradient estimation technique based on an analytic solution, addressing the biased utility problem present in current methods. Using this analytic gradient, we prove that the traditional learning objective suffers from slow local convergence in a classic symmetric auction setting, with the convergence rate degrading quadrati-

cally as the number of bidders increases.

- We propose the *Approximate Best Response Gradient* method, optimizing the *Best Response Distance* objective which provides an upper bound on the approximation quality to the BNE. Our method is proven to achieve local convergence to BNE at a rate independent of the number of bidders and circumvent the slow convergence issue in the symmetric auction setting.

- We conduct extensive experiments across various auction environments, including different auction mechanisms, asymmetric value prior, risk aversion, and alternative gradient estimation approaches. The empirical results show significant improvements in convergence speed and learning efficiency of our method across complex auction scenarios, confirming the efficacy of our proposed method.

## 2. Preliminaries

### 2.1. Auction Game

This paper studies the single-item auction game with $n$ bidders. The value profile $v = (v_1, \ldots, v_n)$ is drawn from a joint prior distribution $\mathcal{F} : \mathbb{R}^n_+ \to [0, 1]$ and each bidder $i \in \{1, \ldots, n\}$ observes a private value realization $v_i$ for the item, where $\mathcal{F}$ is assumed to be common knowledge among all the bidders. Then each bidder $i$ submits a bid $b_i = \beta_i(v_i)$ following his/her bidding strategy $\beta_i : \mathbb{R}_+ \to \mathbb{R}_+$. The auction outcome is determined by a mechanism $\mathcal{M}$, which specifies both the allocation of the item and the payments $p_i$ for each bidder. The risk-neutral *ex-post* utility of each bidder $i$ is calculated as:

$$u_i(v_i, b_i, b_{-i}|\mathcal{M}) = v_i \cdot \mathbb{I}(\text{i wins}) - p_i, \qquad (1)$$

where $\mathbb{I}(\cdot)$ is the indicator function and the index $-i$ denotes the profile of all bidders except for $i$.

For example, under the classic first-price (FP) and second-price (SP) auctions, the *ex-post* utility is formulated as:

$$u_i(v_i, b_i, b_{-i}|\text{FP}) = (v_i - b_i) \cdot \mathbb{I}(b_i > \max_{j \neq i} b_j),$$
$$u_i(v_i, b_i, b_{-i}|\text{SP}) = (v_i - \max_{j \neq i} b_j) \cdot \mathbb{I}(b_i > \max_{j \neq i} b_j). \qquad (2)$$

### 2.2. Bayesian Nash Equilibrium

The Nash Equilibrium is achieved when no one can improve his/her utility by unilaterally changing his/her strategy (Nash, 1950). In the auction game with random player type (*i.e.,* value), NE extends to the Bayesian Nash Equilibrium (Harsanyi, 1967). A typical Bayesian auction game unfolds in three stages: (1) In the *ex-ante* stage, bidders only acquire the value distribution prior; (2) In the *ex-interim* stage, each bidder privately observes his/her value realization; (3) In the *ex-post* stage, the bidding decision is determined. Let

$\bar{u}_i$ be the *ex-interim* utility of bidder $i$:

$$\bar{u}_i(v_i, b_i, \beta_{-i}) = \mathbb{E}_{v_{-i} \sim \mathcal{F}_{-i}(\cdot|v_i)} \left[ u_i(v_i, b_i, \beta_{-i}(v_{-i})) \right], \tag{3}$$

where $\mathcal{F}_{-i}(\cdot|v_i)$ denotes the conditioned distribution of other bidders' value realization $v_{-i}$. Then, bidder $i$'s *ex-ante* utility becomes:

$$U_i(\beta_i, \beta_{-i}) = \mathbb{E}_{v_i \sim \mathcal{F}_i} \left[ \bar{u}_i(v_i, \beta_i(v_i), \beta_{-i}) \right], \tag{4}$$

where $\mathcal{F}_i$ is the marginal distribution of $v_i$.

Following prior studies (Chen & Peng, 2023), the best response of bidder $i$ is computed based on value $v_i$ and others' strategies $\beta_{-i}$ as follows:

$$\text{br}_i(v_i, \beta_{-i}) = \arg\max_b \bar{u}_i(v_i, b, \beta_{-i}). \tag{5}$$

An *ex-interim* $\epsilon$-BNE ($\epsilon \geq 0$) is reached by bidders strategy profile $\beta^* = (\beta_1^*, \ldots, \beta_n^*)$ if, for any bidder $i \leq n$ and value $v_i \in \text{dom } \mathcal{F}_i$:

$$\bar{u}_i(v_i, \beta_i^*(v_i), \beta_{-i}^*) \geq \bar{u}_i(v_i, \text{br}_i(v_i, \beta_{-i}^*), \beta_{-i}^*) - \epsilon. \tag{6}$$

Furthermore, $\beta^*$ is an *ex-ante* $\epsilon$-BNE if, for any bidder $i \leq n$:

$$\begin{aligned}
&\mathbb{E}_{v_i \sim \mathcal{F}_i} \left[ \bar{u}_i(v_i, \beta_i^*(v_i), \beta_{-i}^*)) \right] \\
\geq &\mathbb{E}_{v_i \sim \mathcal{F}_i} \left[ \bar{u}_i(v_i, \text{br}_i(v_i, \beta_{-i}^*), \beta_{-i}^*)) \right] - \epsilon.
\end{aligned} \tag{7}$$

In this paper, we focus primarily on learning the *ex-ante* $\epsilon$-BNE ($\epsilon$-BNE for short). Note that *ex-ante* $\epsilon$-BNE and *ex-interim* $\epsilon$-BNE are equivalent when $\epsilon = 0$, which is simply referred to as BNE.

### 2.3. Gradient-Based Learning

Recent advancements (Bichler et al., 2021; 2023b; Kohring et al., 2023) have investigated the gradient-based learning techniques (Bubeck, 2015; Salimans et al., 2017) to compute BNE in auction games. In these approaches, each bidder's strategy is modeled by a neural network, denoted as $\beta_{\theta_i}(\cdot)$, with parameters $\theta_i$ updated through gradient methods to optimize a specific objective $\mathcal{L}$:

$$\theta_i^{t+1} = \theta_i^t - \alpha \cdot \nabla_{\theta_i} \mathcal{L}(\beta_{\theta_i^t}, \beta_{\theta_{-i}^t}),$$

where $\alpha$ is the learning rate and $\nabla_{\theta_i} \mathcal{L}$ represents the gradient of the objective function w.r.t. the bidder's strategy parameters. This optimization process depends on two key factors: (1) the selection of the learning objective $\mathcal{L}$ and (2) the method for estimating the gradient $\nabla_{\theta_i} \mathcal{L}$.

In practice, most existing methods (Bichler et al., 2021; 2023b; Kohring et al., 2023) unanimously adopt the negative *ex-ante* utility $-U_i(\beta_i, \beta_{-i})$ as the objective $\mathcal{L}$, resulting in

simultaneous gradient ascent on each bidder's utility. The parameter update rule for bidder $i$ can be expressed as:

$$\begin{aligned}
\theta_i^{t+1} &= \theta_i^t + \alpha \cdot \nabla_{\theta_i} U_i(\beta_{\theta_i^t}, \beta_{\theta_{-i}^t}) \\
&= \theta_i^t + \alpha \cdot \mathbb{E}_{v_i} \left[ \nabla_{\theta_i} \bar{u}_i(v_i, \beta_{\theta_i^t}(v_i), \beta_{\theta_{-i}^t}) \right],
\end{aligned} \tag{8}$$

where various approximation methods (Bichler et al., 2021; Kohring et al., 2023) have been proposed to estimate the *ex-interim* utility's gradient $\nabla_{\theta_i} \bar{u}_i$. These methods enable practical implementations of learning BNE in complex auction environments.

## 3. Gradient Estimation

The need to approximate the utility's gradient $\nabla_{\theta_i} \bar{u}_i$ arises from the discontinuous nature of the *ex-post* utility function, which includes a non-smooth indicator operator $\mathbb{I}(\cdot)$ defined in Eq. (2). As shown in previous work (Kohring et al., 2023), such discontinuities can lead to inaccurate gradient estimation for the Monte Carlo (MC) method when calculating the *ex-interim* utility's gradient $\nabla_{\theta_i} \bar{u}_i(v_i, \theta_i)$ (we rewrite $\bar{u}_i(v_i, \theta_i) = \bar{u}_i(v_i, \beta_{\theta_i}(v_i), \beta_{-i})$ for brevity):

$$\begin{aligned}
\nabla_{\theta_i} \bar{u}_i(v_i, \theta_i) &= \nabla_{\theta_i} \mathbb{E}_{v_{-i}|v_i} \left[ u_i(v_i, \beta_{\theta_i}(v_i), \beta_{-i}(v_{-i})) \right] \\
&\not\approx \frac{1}{K} \sum_{j=1}^{K} \nabla_{\theta_i} u_i(v_i, \beta_{\theta_i}(v_i), \beta_{-i}(v_{-i}^j))
\end{aligned} \tag{9}$$

where $K$ denotes the sample size. As a result, Bichler et al. (2021) reported that strategies learned using MC estimation on gradient tend to converge to zero bidding in FP auctions. To illustrate this issue, we exemplify the FP auction with the *ex-post* utility defined in Eq. (2), and compute the MC estimation on gradient $\nabla_{\theta_i}^{\textbf{MCg}} \bar{u}_i$ of bidder $i$ as follows:

$$\begin{aligned}
\nabla_{\theta_i}^{\textbf{MCg}} \bar{u}_i(v_i, \theta_i) &= \frac{1}{K} \sum_{j=1}^{K} \nabla_{\theta_i} u_i(v_i, \beta_{\theta_i}(v_i), \beta_{-i}(v_{-i}^j)) \\
&\approx -\Pr(\text{i wins}) \cdot \nabla_{\theta_i} \beta_{\theta_i}(v_i).
\end{aligned} \tag{10}$$

where $\Pr(\text{i wins}) = \mathbb{E}_{b_{-i}|v_i} \mathbb{I}(\beta_{\theta_i}(v_i) > \max\{b_{-i}\})$. The key issue here is that the MC-estimated gradient's coefficient for the bidder $i$'s bidding network $\nabla_{\theta_i} \beta_{\theta_i}(v_i)$ is consistently negative unless the bidder places a minimal bid. This implies that the estimated gradient persistently encourages the bidder to lower their bid until $\beta_{\theta_i}(v_i) = 0$, which explains the zero-bidding problem in MC gradient estimation.

### 3.1. Existing Gradient Estimation

To address this issue, existing works (Bichler et al., 2021; Li & Wellman, 2021; Kohring et al., 2023) turn to utilize different gradient estimation methods to approximate the *ex-interim* utility's gradient. Early works adopt black-box optimization techniques, such as Evolution Strategies (ES),

to estimate the pseudo-gradient as an approximation of the utility's gradient (Bichler et al., 2021; Li & Wellman, 2021). Specifically, the ES algorithm approximates the *ex-interim* utility with a Gaussian-smoothed utility function, where the bidding strategy's parameter $\theta_i$ is perturbed with Gaussian noise, then the gradient of this smoothed utility can be estimated as follows:

$$\nabla_{\theta_i}^{\mathbf{ES}}\bar{u}_i(v_i;\theta_i) \approx \nabla_{\theta_i}\mathbb{E}_{\epsilon\sim\mathcal{N}(0,\sigma^2 I)}[\bar{u}_i(v_i;\theta_i + \epsilon)]$$
$$= \mathbb{E}_{\epsilon\sim\mathcal{N}(0,\sigma^2 I)}[\frac{\epsilon}{\sigma^2}\bar{u}_i(v_i;\theta_i + \epsilon)].$$

In addition to perturbing the parameters of the bidding function, this approach can also be applied to perturb the outputs of the bidding function (*i.e.*, bid values). This alternative method, known as the REINFORCE algorithm (Salimans et al., 2017; Kohring et al., 2023), perturbs the bid function $\beta_i$ into a mixed strategy such that $b_i \sim \mathcal{N}(\beta_{\theta_i}(v_i), \sigma^2 I)$. Similarly, the gradient can be estimated as:

$$\nabla_{\theta_i}^{\mathbf{RE}}\bar{u}_i(v_i, \beta_{\theta_i}(v_i), \beta_{-i})$$
$$\approx \nabla_{\theta_i}\mathbb{E}_{\epsilon\sim\mathcal{N}(0,\sigma^2 I)}[\bar{u}_i(v_i, \beta_{\theta_i}(v_i) + \epsilon, \beta_{-i})]$$
$$= \mathbb{E}_{\epsilon\sim\mathcal{N}(0,\sigma^2 I)}[\frac{\epsilon}{\sigma^2}\bar{u}_i(v_i, \beta_{\theta_i}(v_i) + \epsilon, \beta_{-i})\nabla_{\theta_i}\beta_{\theta_i}(v_i)].$$

Despite the widely recognized effectiveness in black-box optimization, the ES algorithm faces criticism for the high variance and computational complexity due to the perturbation-based estimation (Bichler et al., 2023a; Kohring et al., 2023). To alleviate this concern, Smooth Market (SM) (Kohring et al., 2023) introduced a smooth *ex-post* utility function that approximates the original discontinuous utility by substituting the winning indicator $\mathbb{I}(i \text{ wins})$ and distributing the payment with a smooth soft-max operator:

$$\nabla_{\theta_i}^{\mathbf{SM}}\bar{u}_i(v_i, \theta_i) \approx \mathbb{E}_{v_{-i}|v_i}[\nabla_{\theta_i}u_i^{\mathrm{SM}}(v_i, \beta_{\theta_i}(v_i), \beta_{-i}(v_{-i}))],$$

where $u_i^{\mathrm{SM}}(v_i, b_i, b_{-i}) = (v_i - p_i^{\mathrm{SM}})\cdot\frac{\exp(b_i/\lambda)}{\sum_{j=1}^n\exp(b_j/\lambda)}$. Thus the approximated utility function is fully differentiable, making the first-order optimization applicable: the gradient can be efficiently estimated via sampling and back-propagation, without relying on noisy perturbations in zeroth-order methods like ES and REINFORCE. This approach improves both approximation quality and computational efficiency compared to previous methods (Kohring et al., 2023).

## 3.2. Proposed Gradient Estimation

Although the aforementioned approaches can estimate the *ex-interim* utility's gradient, the utility function under their estimation is modified by either Gaussian noise (*e.g.*, $\nabla_{\theta_i}^{\mathbf{ES}}\bar{u}_i, \nabla_{\theta_i}^{\mathbf{RE}}\bar{u}_i$) or softmax approximation (*e.g.*, $\nabla_{\theta_i}^{\mathbf{SM}}\bar{u}_i$), which can introduce error in the final learned strategy (Kohring et al., 2023). Surprisingly, we find that the gradient of *ex-interim* utility $\nabla_{\theta_i}\bar{u}_i$ can be directly estimated via Monte Carlo sampling in certain mechanisms (*e.g.*, FP, SP),

*Table 1.* Comparison of existing gradient estimation methods.

| Gradient Estimation Approach | Optimization Method | Introduced Utility Bias |
|---|---|---|
| Evolution Strategy $\nabla^{\mathbf{ES}}$ | zeroth-order | Gaussian Noise |
| REINFORCE $\nabla^{\mathbf{RE}}$ | zeroth-order | Gaussian Noise |
| Smooth Market $\nabla^{\mathbf{SM}}$ | first-order | Softmax |
| The Derived Eq.(11) | first-order | Free |

by leveraging an analytic gradient solution. As elucidated in Tab. 1, this estimation avoids the model bias of existing methods by preserving the original utility function. In particular, we present this analytic gradient computation under FP and SP auctions in Equation (11) (See Appendix C for detailed proofs and additional scenarios):

$$\nabla_{\theta_i}\bar{u}_i(v_i, \beta_{\theta_i}(v_i), \beta_{-i}) = \frac{\partial}{\partial b_i}\bar{u}_i(v_i, b_i, \beta_{-i})\cdot\nabla_{\theta_i}\beta_{\theta_i}(v_i)$$
$$= \begin{cases} [(v_i - b_i)f_{m_i}(b_i) - F_{m_i}(b_i)]\cdot\nabla_{\theta_i}\beta_{\theta_i}(v_i), & \text{FP} \\ (v_i - b_i)f_{m_i}(b_i)\cdot\nabla_{\theta_i}\beta_{\theta_i}(v_i), & \text{SP} \end{cases}$$
$$(11)$$

Here, $f_{m_i}$ and $F_{m_i}$ are the probability density function (pdf) and cumulative density function (cdf), respectively. Both of them can be estimated via sampling market prices $\max\{\beta_{-i}(v_{-i})\}$ with $v_{-i} \sim \mathcal{F}_{-i}(\cdot|v_i)$ given $v_i$: the cdf can be naturally estimated via MC estimation and the pdf can be estimated via kernel density estimation (KDE) or using a histogram of sampled market prices. Therefore, the utility gradient $\nabla_{\theta_i}\bar{u}_i$ can be correctly approximated by the Monte Carlo estimated distribution, enabling learning with first-order methods based on this analytic gradient computation. We name this method as *MC on distribution* to differentiate it from the *MC on gradient* one in Eq. (10).

Moreover, this analytic gradient computation allows for investigating the convergence ability of gradient methods to BNE strategies. We follow a classic symmetric auction setting (Feng et al., 2021) with shared linear bidding strategies:

**Assumption 3.1.** Consider a symmetric $n$-bidder first-price auction with uniform prior $\mathcal{F}_i = U[0,1]$. Bidders share a linear bidding function $\beta_{\theta_i}(v_i) = wv_i + c$ with non-negative parameter $\theta_i := (w, v)^\intercal \in R_+^2$ (to ensure bids $b_i \geq 0$).

The symmetric BNE solution to this auction game is $\beta_i^*(v_i) = \frac{n-1}{n}$ (*i.e.*, $\theta_i^* = (\frac{n-1}{n}, 0)^\intercal$). Consistent with previous studies (Kohring et al., 2023; Bichler et al., 2021), we focus on maximizing the *ex-ante* utility via gradient ascent, herein referred to as the *Utility Gradient* method:

$$\theta_i^{t+1} \leftarrow \theta_i^t + \alpha\nabla_{\theta_i}U_i(\beta_{\theta_i^t}, \beta_{\theta_{-i}^t}). \quad \text{(Utility Gradient)}$$

Then we can prove that our derived gradient solution in Eq. (11) converges to the BNE while the MC estimation on gradient in Eq. (10) incorrectly converges to zero bidding:

**Proposition 3.2** (Convergence Ability). *Under Assump. 3.1, projected gradient ascent[1] via utility gradient satisfies:*

---

[1] We use projected gradient ascent $\theta_i^{t+1} \leftarrow \text{Proj}(\theta_i^t + \alpha\nabla_{\theta_i}U_i)$

1. *Learning with the derived gradient in Eq.* (11) *converges to the symmetric BNE:* $\beta^*_{\theta_i}(v_i) = \frac{n-1}{n} v_i$.
2. *Learning with MC estimation on gradient in Eq.* (10) *converges to the zero-bidding strategy:* $\beta_{\theta_i}(v_i) = 0$.

## 4. Proposed Learning Algorithm

### 4.1. Convergence Rate of Utility Gradient

Despite the convergence guarantee in Proposition 3.2, we find the utility gradient method suffers from slow convergence in the vicinity of the BNE point $\theta^*_i$ when the bidder number $n$ increases. To demonstrate this phenomenon, we conduct numerical experiments under Assump. 3.1 by initializing the parameters $\theta^0_i$ in the neighborhood of the BNE point $\theta^*_i = (\frac{n-1}{n}, 0)^\intercal$. As depicted in Fig. 2, the utility gradient method fails to converge to the BNE point $\theta^*$ within limited training steps if the bidder number $n$ grows from 2 to 10. Such poor convergence results persist even if we extend the training times twice, labeled as "Utility Grad (+300 steps)" in the figure.

We identify that such a slow convergence phenomenon is raised due to the local property of the utility objective: $U_i(\beta_{\theta_i}, \beta_{\theta_{-i}})$. To capture the vicinal property of the BNE point $\theta^*_i$, we define a neighbor region with $\Theta(\frac{1}{n})$ radius:

$$\mathcal{N}(\theta^*_i) := \{\theta_i \in R^2_+ \mid \|\theta_i - \theta^*_i\| \leq \frac{1}{2n}\}.$$

By examining the Hessian matrix $\boldsymbol{H}(\theta_i)$ of the objective $U_i$ in this neighborhood, we find:

**Lemma 4.1.** *Under Assump. 3.1, the Hessian matrix $\boldsymbol{H}(\theta_i)$ of $U_i(\beta_{\theta_i}, \beta_{\theta_{-i}})$ in $\mathcal{N}(\theta^*_i)$ has a condition number in the order of $\Theta(n^2)$.*

In optimization, the convergence speed of gradient-based methods depends on the condition number of the objective's Hessian; higher condition numbers generally lead to slower convergence (Bubeck, 2015). In our auction setting, the utility function's Hessian has a condition number growing quadratically as $n$. This ill-conditioned property (Tong et al., 2021) explains the slow convergence observed for large $n$.

Intuitively, for a bidder who should submit a low bid when holding a low valuation, his/her chance of winning becomes negligible as $n$ increases. Consequently, the utility gain of deviating from the current strategy $\beta_i$ to the BNE strategy $\beta^*_i$ also diminishes with increasing $n$. This results in vanishing gradient magnitudes near the BNE for low valuations. Conversely, the bidder's utility gradient remains significant when their valuation is high and does not degenerate with increasing $n$. This imbalance in gradient magnitudes, caused by the vanishing gradient at low valuations, can lead to slower convergence rates during optimization.

to ensure non-negative parameter, where the projection operator is: $\mathrm{Proj}(w, c) = (\max\{w, 0\}, \max\{c, 0\})^\intercal$.

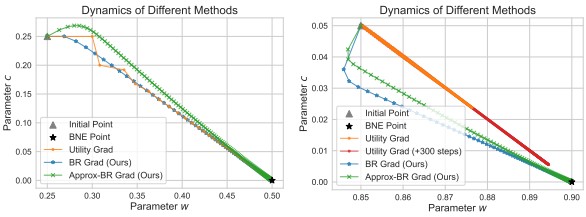

(a) Dynamics when $n = 2$     (b) Dynamics when $n = 10$

*Figure 2.* Learning dynamics of different methods under Assumption 3.1. The initial point $\theta^0_i$ is picked at the vicinity of $\theta^*_i$, with $\theta^0_i = \theta^*_i + (\frac{-1}{2n}, \frac{1}{2n})^\intercal$. All methods run 150 steps by default and we plot the best-performing results after learning-rate grid search.

Formally, we prove that the local convergence rate satisfies:

**Theorem 4.2** (Utility Gradient Converge Rate). *Under Assump. 3.1, for projected gradient ascent via utility gradient:*
1. *It converges to the BNE $\theta^*_i$ at a local rate of $\mathcal{O}(n^2 \log \frac{1}{d})$, where $d$ is the last-iterate distance to BNE $\|\theta^t_i - \theta^*_i\|$.*
2. *Let $\theta^0_i = (\frac{n-2}{n}, \frac{1}{n+1})^\intercal$, it will require $\Omega(n^2 \log \frac{1}{d})$ iterations to reach $\|\theta^t_i - \theta^*_i\| \leq d$.*

These results indicate that the number of iterations required to achieve a specified accuracy increases quadratically with the number of bidders $n$, highlighting the inefficiency of the utility gradient method. Moreover, this slow-converging problem still exists when applied with accelerated gradient methods, such as the Momentum (Kingma & Ba, 2015) or Nesterov method (Beck & Teboulle, 2009), which we further discuss in Appendix C.2.3 and show that they also suffer from growing complexity with bidder number $n$. In contrast, in the following sections, we will see that our proposed *BR gradient* method achieves a convergence rate of $\mathcal{O}(\log \frac{1}{d})$, proved to avoid the $n$ term.

### 4.2. Best-Response Distance Objective

The slow convergence of the utility objective encourages us to explore a more robust objective in auction games, which can directly capture the strategy-wise difference of $\beta_i$ and $\beta^*_i$ so that the convergence rate won't degenerate when $n$ increases. Inspired by recent success of equilibrium-aware loss in learning equilibrium (Gemp et al., 2024; Huang et al., 2024), we propose a new objective called *Best Response Distance* (BR Distance):

$$\mathcal{L}_{\mathrm{BR}}(\beta_\theta) = \sum_{i \leq n} \frac{1}{2} \mathbb{E}_{v_i} \| \beta_{\theta_i}(v_i) - \mathrm{br}_i(v_i, \beta_{\theta_{-i}}) \|^2 \quad (12)$$

where $\mathrm{br}_i(v_i, \beta_{\theta_{-i}}) = \arg\max_b \bar{u}_i(v_i, b, \beta_{\theta_{-i}})$.

By directly measuring the distance between current strategy $\beta_{\theta_i}(\cdot)$ with the best-response bid $\mathrm{br}(\cdot, \beta_{-i})$ (Armantier et al., 2008), this loss sidesteps the utility objective's gradient vanishing problem when bidder numbers increase.

Moreover, assuming the *ex-interim* utility $\bar{u}_i(v_i, b_i, \beta_{-i})$ to be $L$-Lipschitz continuous in $b_i$, the BR distance can serve as an upper bound of the approximation factor $\epsilon$ in $\epsilon$-BNE:

**Lemma 4.3.** *With an $L$-Lipschitz continuous ex-interim utility, the strategy profile $\hat{\beta}$ forms an $\epsilon$-BNE such that $\epsilon \leq \sqrt{2\mathcal{L}_{\mathrm{BR}}(\hat{\beta}) \cdot L}$.*

Therefore a lower BR distance leads to a closer approximation to the BNE. Like utility gradient method, we can optimize this objective by applying gradient update for each bidder's strategy $\theta_i$, dubbed as the *BR Gradient* method:

$$\theta_i^{t+1} \leftarrow \theta_i^t - \alpha g_{\mathrm{br}_i}(\beta_{\theta^t}), \qquad \text{(BR Gradient)}$$

where the gradient is calculated as:

$$g_{\mathrm{br}_i}(\beta_{\theta^t}) = \mathbb{E}_{v_i}\left[\left(\beta_{\theta_i^t}(v_i) - \mathrm{br}_i(v_i, \beta_{\theta_{-i}^t})\right)\cdot\nabla_{\theta_i}\beta_{\theta_i^t}(v_i)\right]. \tag{13}$$

Furthermore, we've derived the following convergence rate in comparison to the utility gradient method:

**Theorem 4.4** (BR Gradient Converge Rate). *Under Assump. 3.1, projected gradient descent via BR gradient converges to the BNE $\theta_i^* = (\frac{n-1}{n}, 0)^\intercal$ at a rate of $\mathcal{O}(\log\frac{1}{d})$.*

With an $\mathcal{O}(\log\frac{1}{d})$ convergence rate, the BR gradient method is proved to fundamentally avoid the growing complexity with increasing $n$. As illustrated in Fig. 2, the BR gradient dynamics converge to the BNE point within limited steps, even for a large number of bidders $n$.

### 4.3. Learning via Approximate BR Gradient

Although the BR Gradient is proven to avoid the slow converging problem, it requires an arg-max operator over all bids: $\mathrm{br}_i(v_i, \beta_{-i}) = \arg\max_b \bar{u}_i(v_i, b, \beta_{-i})$, which can introduce vast computational burdens in practice. To circumvent this issue, we propose to utilize a locally approximated best response $\widehat{\mathrm{br}}_i$ instead of directly computing the arg-max.

Firstly, we consider to utilize the local second-order Taylor's expansion of the *ex-interim* utility function (we denote $\bar{u}_i(b) = \bar{u}_i(v_i, b, \beta_{-i})$ and $b_i = \beta_{\theta_i}(v_i)$ for clarity):

$$\bar{u}_i(b) \approx \bar{u}_i(b_i) + \frac{\partial \bar{u}_i(b_i)}{\partial b_i}(b - b_i) + \frac{1}{2}\frac{\partial^2 \bar{u}_i(b_i)}{\partial b_i^2}(b - b_i)^2.$$

If the *ex-interim* utility is concave locally, *i.e.*, $\partial^2\bar{u}_i/\partial b_i^2 < 0$, the maximum of the quadratic approximation provides an estimate of the best response, resulting in a *second-order approximate best repsonse*:

$$\widehat{\mathrm{br}}_i^{\,2\mathrm{nd}} \approx b_i - \left(\partial^2\bar{u}_i/\partial b_i^2\right)^{-1} \cdot \partial\bar{u}_i/\partial b_i. \tag{14}$$

This formula resembles Newton's method (Nesterov, 2018) but is used here to approximate the local best response

rather than to optimize the entire objective function. Additionally, the second-order gradient of $\partial^2\bar{u}_i/\partial b_i^2$ can also be calculated analytically, similar to the first-order gradient in Eq. (11). Specifically, in FP or SP auctions:

$$\frac{\partial^2 \bar{u}_i}{\partial b_i^2} = \begin{cases} (v_i - b_i)f_{m_i}(b_i) - 2f'_{m_i}(b_i), & \text{FP} \\ (v_i - b_i)f_{m_i}(b_i) - f'_{m_i}(b_i), & \text{SP} \end{cases} \tag{15}$$

where $f'_{m_i}(\cdot)$ is the gradient of market price's pdf. To estimate the pdf's gradient, we can use any differentiable estimation method, such as KDE with a differentiable kernel, or other fully differentiable methods like SM[2].

On the other hand, if the *ex-interim* utility is locally convex, *i.e.*, $\partial^2\bar{u}_i/\partial b_i^2 \geq 0$, the quadratic approximation has unbounded maxima. To address this, we replace the second-order gradient with a negative constant to ensure the update direction improves the utility, resulting in a *first-order approximate best response*:

$$\bar{u}_i(b) \approx \bar{u}_i(b_i) + \frac{\partial \bar{u}_i(b_i)}{\partial b_i}(b - b_i) - \frac{1}{2\gamma}(b - b_i)^2$$
$$\Rightarrow \widehat{\mathrm{br}}_i^{\,1\mathrm{st}} \approx b_i + \gamma \cdot \partial\bar{u}_i/\partial b_i \tag{16}$$

With these approximations, we have the following *Approximate BR Gradient* method:

$$\theta_i^{t+1} \leftarrow \theta_i^t - \alpha\hat{g}_{\mathrm{br}_i}(\beta_{\theta^t}) \qquad \text{(Approx BR Gradient)}$$
$$\text{where } \hat{g}_{\mathrm{br}_i}(\beta_{\theta^t}) = \mathbb{E}_{v_i}[(\beta_{\theta_i^t}(v_i) - \widehat{\mathrm{br}}_i) \cdot \nabla_{\theta_i}\beta_{\theta_i^t}(v_i)],$$

and the approximate best response $\widehat{\mathrm{br}}_i$ is calculated via Eq. (14) or Eq. (16) depending on the local condition, which we've detailed in Algorithm 1 in Appendix E.2.

In fact, when applying the first-order approximation, our method reduces to the previous utility gradient method:

$$-\hat{g}_{\mathrm{br}_i}(\beta_\theta) = -\mathbb{E}_{v_i}[(\beta_{\theta_i}(v_i) - \widehat{\mathrm{br}}_i^{\,1\mathrm{st}}) \cdot \nabla_{\theta_i}\beta_{\theta_i}(v_i)]$$
$$= \gamma\nabla_{\theta_i}\bar{u}_i(v_i, \beta_{\theta_i}(v_i), \beta_{-i}).$$

But in the vicinity of the BNE, where the slow convergence problem of utility gradient happens, the bid $b_i$ is close to the best response. Using the optimality condition of best response $\mathrm{br}_i = \arg\max_{b_i} \bar{u}_i(v_i, b_i, \beta_{-i})$, we have:

$$\partial\bar{u}_i/\partial b_i = 0, \partial^2\bar{u}_i/\partial b_i^2 < 0. \tag{17}$$

Such concavity validates the use of a second-order approximation for the best response $\widehat{\mathrm{br}}_i^{\,2\mathrm{nd}}$ in the neighborhood of BNE. Consequently, the local convergence properties differ from those of the utility gradient. Specifically, we've established the following local convergence result:

---

[2]The contribution of optimization objective is orthogonal to gradient estimation, allowing for various estimation methods.

**Theorem 4.5** (Approximate BR Gradient Converge Rate). *Under Assump. 3.1, projected gradient descent via approximate BR gradient locally converges to the BNE $\theta_i^* = (\frac{n-1}{n}, 0)^\intercal$ at a rate of $\mathcal{O}(\log \frac{1}{d})$.*

Like the original BR gradient, the approximate BR gradient method also achieves an $\mathcal{O}(\log \frac{1}{d})$ local convergence rate, thereby avoiding the locally slow-converging problem of the utility gradient. As depicted in Fig. 2, the learned strategy using approximate BR gradient also successfully converges to the BNE point even with increased bidder numbers.

# 5. Experiments

## 5.1. Implementation Details

We conduct the experiments in a more general setting where the bidding function is a 3-layer MLP instead of the linear model in Assumption 3.1 and evaluate performance across different mechanisms, asymmetric auctions, risk-averse utilities, and alternative gradient estimation approaches. To assess the learned bidding strategies and their approximation of the BNE, we analyze several classic auction games with known BNE solutions and measure the L2-distance between the learned strategy $\beta_{\theta_i}(\cdot)$ and the BNE solution $\beta_i^*(\cdot)$ by sampling $K$ values $v_i^j$ from the prior $\mathcal{F}_i$:

$$L_{2_i} = \Big( \frac{1}{K} \sum_{j=1}^{K} \big( \beta_{\theta_i}(v_i^j) - \beta_i^*(v_i^j) \big)^2 \Big)^{1/2},$$

where the BNE solutions are detailed in Appendix E.1. For comparisons, we primarily utilize the following three methods to learn BNE in different auction game settings:

- SM (Kohring et al., 2023): the existing state-of-the-art method for learning BNE strategies, which maximizes $U_i$ with soft-max approximated gradient $\nabla_{\theta_i}^{\mathbf{SM}} \bar{u}_i$.
- Utility Grad: the utility gradient method using our derived gradient solution in Eq. (11) to maximize $U_i$.
- Approx BR: the proposed approximate BR gradient method (or Approx BR for short), detailed in Algorithm 1.

Following (Kohring et al., 2023; Bichler et al., 2021), we use a shared MLP network for bidders with the same value prior and run each method with 2,000 steps. More details can be found in Appendix E.

## 5.2. Main Results

We benchmark the L2 distance in the symmetric auction games with a uniform prior under first-price and second-price mechanisms like previous work (Bichler et al., 2021; Kohring et al., 2023; Huang et al., 2024), where the BNE solutions are $\beta_i^*(v_i) = \frac{n-1}{n} v_i$ and $\beta_i^*(v_i) = v_i$ respectively. We increase the bidder numbers $n$ and validate the convergence improvement of our proposed method in Tab. 2, where we report the averaged L2 results across 5 initializations.

*Table 2.* Learned strategies' L2 distance to BNE using different methods in FP/SP symmetric auctions with varying numbers of bidders $n$. We report the averaged L2 across 5 initializations after a hyper-parameter search for every method. The L2 values are evaluated at different steps ("@steps") during the training stage.

| $\mathcal{M}$ | n | Algorithm | L2@500 | L2@1k | L2@2k | t/iter |
|---|---|---|---|---|---|---|
| FP | 2 | SM | **4.12e-3** | **3.75e-3** | 3.08e-3 | 0.013s |
| | | **Utility Grad** | 5.35e-3 | 4.75e-3 | **2.82e-3** | 0.002s |
| | | Approx BR | 5.46e-3 | 5.42e-3 | 3.01e-3 | 0.002s |
| | 5 | SM | 3.74e-2 | 1.83e-2 | 1.57e-2 | 0.035s |
| | | Utility Grad | 3.49e-2 | 1.62e-2 | 1.51e-2 | 0.002s |
| | | **Approx BR** | **1.35e-2** | **7.77e-3** | **5.20e-3** | 0.002s |
| | 10 | SM | 1.05e-1 | 7.39e-2 | 5.01e-2 | 0.091s |
| | | Utility Grad | 8.84e-2 | 6.94e-2 | 5.44e-2 | 0.002s |
| | | **Approx BR** | **6.21e-2** | **1.28e-2** | **9.65e-3** | 0.002s |
| SP | 2 | SM | **5.26e-3** | **4.53e-3** | **3.57e-3** | 0.012s |
| | | Utility Grad | 5.31e-3 | 4.61e-3 | 3.73e-3 | 0.002s |
| | | Approx BR | 5.31e-3 | 4.62e-3 | 3.77e-3 | 0.002s |
| | 5 | SM | 3.40e-2 | 1.89e-2 | 1.73e-2 | 0.035s |
| | | Utility Grad | 2.91e-2 | 1.59e-2 | 1.34e-2 | 0.002s |
| | | **Approx BR** | **7.83e-3** | **4.89e-3** | **3.45e-3** | 0.002s |
| | 10 | SM | 1.40e-1 | 7.57e-2 | 3.51e-2 | 0.090s |
| | | Utility Grad | 9.35e-2 | 5.35e-2 | 3.17e-2 | 0.002s |
| | | **Approx BR** | **1.33e-2** | **5.91e-3** | **3.73e-3** | 0.002s |

As shown in Tab. 2, we find that the proposed approximate BR gradient method significantly enhances convergence speed as the number of bidders increases. While all methods achieve near-optimal solutions with L2 distances at the order of 1e-3 when the number of bidders is small ($n = 2$), differences become pronounced as $n$ grows. As the number of bidders $n$ increases, the SM and Utility Grad methods exhibit slow convergence, resulting in final L2 distances of approximately 5e-2 and 3e-2 when $n = 10$. In contrast, the Approx BR method consistently delivers solutions with L2 distances at the 1e-3 order. This reduction in L2 distance indicates that the strategies learned by our method produce a more accurate approximation of the BNE.

For a clearer understanding of different L2 magnitudes, Fig. 5b provides a visual comparison of the strategies, where the "SM+aBR" strategy (ours) closely aligns with the BNE solution, while the "SM" strategy exhibits a noticeable deviation from the BNE. Specifically, the "SM+aBR" strategy achieves an L2 distance of approximately 5e-3, in contrast to the "SM" strategy, which reaches an L2 distance of about 7e-2. Referring back to the results in Tab. 2, where our Approx BR method consistently yields average L2 distances at the order of 1e-3, this confirms the precision of the strategies learned through our method in approximating the BNE.

Additionally, we explain the substantial reduction in the time required for each step in the Utility Grad and Approx BR methods, as shown in the last column (t/iter) of Tab. 2. The reason is that both methods estimate gradients based

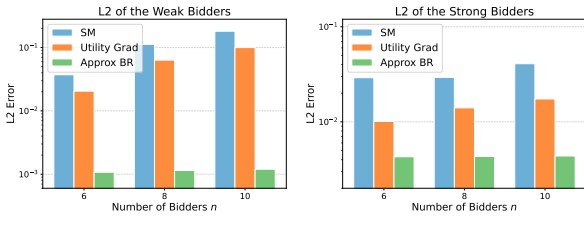

(a) L2 of the weak bidders      (b) L2 of the strong bidders

*Figure 3.* L2 error of learned strategies under the SP asymmetric auction with 2 different (weak/strong) value priors.

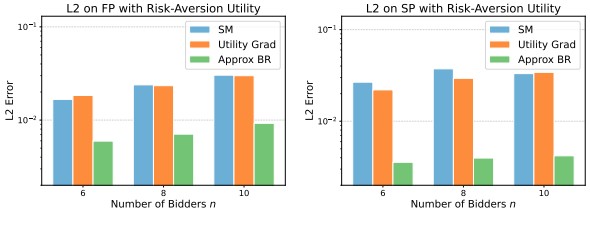

(a) FP with risk aversion      (b) SP with risk aversion

*Figure 4.* L2 error of learned strategies under the FP/SP symmetric auction settings with risk-averse bidders

on the proposed MC on distribution estimation with directly computed solution in (11), whereas SM relies on sampling the *ex-post* utility and approximating the gradient through backpropagation. The direct estimation significantly improves the efficiency of our gradient estimation method.

### 5.3. Further Extensions

**Extension to Asymmetric Auctions.** We also explore the asymmetric auction setting, where bidders have different value priors. Following previous work on learning asymmetric auction games (Bichler et al., 2023a), we consider the FP auction with 2 bidders whose value priors are $U[0, 0.5]$ (the weak bidder) and $U[0, 1]$ (the strong bidder). As depicted in Fig. 11 in Appendix E.1, the bidding strategies learned via our Approx BR method closely align with the BNE strategy, confirming convergence even in this asymmetric setting.

We then scale up the number of bidders to evaluate convergence speed with a similar prior setting in SP auctions, where half of the bidders belong to the weak type $U[0, 0.5]$ and the rest strong type follows $U[0, 1]$. As shown in Fig. 3, both SM and Utility Grad methods exhibit L2 distances above 1e-2, while our proposed Approx BR method consistently achieves precise solutions with L2 distances at the order of 1e-3 for both types of bidders, demonstrating a much closer approximation to the BNE. In addition, the Utility Grad also outperforms the SM's prediction, highlighting the effectiveness of our gradient estimation approach.

**Extension to Risk Aversion.** In practical trading scenarios, the bidders might be risk-averse, which means their utility may not be identical to the monetary payoff. To address this, we extend our experiments to cases where each bidder has a Von Neumann–Morgenstern utility function (Krishna, 2010). Similar to prior works (Bichler et al., 2021; 2023a), we assume the risk-averse bidders possess the utility function defined as the payoff raised to the power of $\rho < 1$. By measuring the L2 distance to analytic BNE solutions, we can evaluate the convergence rates of the three methods in the risk-averse setting. As depicted in Fig. 4, our proposed Approx BR continuously learns precise solutions with the L2 distance at the order of 1e-3 for both types of bidders, outperforming other methods by a clear margin.

**Alternative Gradient Estimations.** The proposed approximate BR gradient method is not limited to a specific gradient estimation technique. To demonstrate its adaptability, we incorporate the SM gradient estimation into the approximate BR (aBR for short) gradient learning framework, referring to this combination as "SM+aBR". Given that SM exhibits slow convergence with large bidder numbers $n$, we benchmark both SM and the aBR-augmented version in the 10-bidder FP auction setting of Tab. 2 to examine whether our proposed method can enhance convergence speed.

We run both methods with five initializations and plot the L2 distance over training time in Fig. 5a. As illustrated, after applying the approximate BR gradient method from the 200-th step, the L2 distance decreases significantly faster than the original SM method and quickly converges around 1,000 steps. Moreover, we present the best-performing learned strategies across 5 initialization for both methods in Fig. 5b. The results reveal that the strategy learned by SM shows a noticeable distance from the BNE solution, whereas the learned strategy of SM+aBR aligns closely with the BNE strategy. This confirms the flexibility of the proposed Approx BR method to effectively utilize different gradient estimation techniques without the analytic gradients.

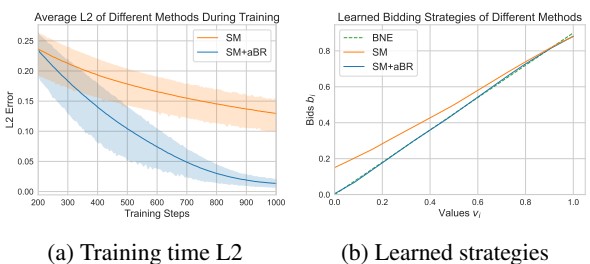

(a) Training time L2      (b) Learned strategies

*Figure 5.* Learning results of the SM method (Kohring et al., 2023) and the augmented version with our proposed Approx BR method (SM+aBR) under the 10-bidder symmetric FP auction.

## 6. Conclusion

In this paper, we introduce the *Approximate Best Response Gradient* method, a new approach for efficiently learning Bayesian Nash Equilibrium in auction games. By utilizing an analytic solution for utility gradient estimation, we

not only improve the efficiency of the gradient computation but also enable theoretical convergence analysis. Our method optimizes the proposed *Best Response Distance* objective, which achieves a significantly faster local convergence rate of $\mathcal{O}(\log \frac{1}{d})$ in a classic symmetric auction setting, compared to the traditional utility gradient method's $\mathcal{O}(n^2 \log \frac{1}{d})$ rate. Extensive experiments validate the superior convergence speed of our approach across various auction scenarios, including different auction mechanisms, asymmetric value distributions, risk-averse utilities, and alternative gradient estimation techniques.

## Acknowledgments

This research is supported by the National Natural Science Foundation of China (92270114). This research was also supported by the advanced computing resources provided by the Supercomputing Center of the USTC.

## Impact Statement

This paper presents work whose goal is to advance the field of Machine Learning. There are many potential societal consequences of our work, none which we feel must be specifically highlighted here.

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

# A. Related Work

Our work is mainly related to methods for solving equilibrium in auction games, which has been addressed via various approaches.

**Numerical Equilibrium Solver**: Computing Nash Equilibrium (NE) is known for its computational complexity, with even the two-player (aka bidder) bi-matrix game problem being PPAD-complete (Chen et al., 2009). This complexity extends to auction settings, where solving BNE is also PPAD-complete in both continuous (Filos-Ratsikas et al., 2021) and discrete value scenarios (Filos-Ratsikas et al., 2024; Chen & Peng, 2023). Despite this complexity, there have been positive developments in numerical methods for solving BNE in specific auction contexts. Marshall et al. (1994) proposed a numerical method for solving BNE in first-price auctions, named backward-shooting, which has since been applied to both continuous (Bajari, 2001; Gayle & Richard, 2008) and discrete settings (Wang et al., 2020). Other numerical methods include polynomial approximations (Bajari, 2001; Hubbard & Paarsch, 2009) and best-response dynamics (Bosshard et al., 2020).

**Learning BNE in Discrete Auction Games**: Recent advancements have investigated the application of machine learning techniques to compute the BNE in both discrete and continuous auction games. In discrete value scenarios, initial approaches applied no-regret learning algorithms, known to converge to NE in zero-sum normal-form games (Freund & Schapire, 1999; Lei et al., 2021), to auction scenarios (Hartline et al., 2015). Theoretical results show that no-regret learning can converge to BNE in several simplified auction settings (Feng et al., 2021), while it might fail to converge in auctions with fixed values (Deng et al., 2022). Researchers have also introduced other agent-based learning methods (e.g. reinforcement learning) into discrete auctions (Feng et al., 2021; Kolumbus & Nisan, 2022; Banchio & Skrzypacz, 2022). More advanced methods include the application of online convex optimization techniques (Bichler et al., 2023a) and the development of unified frameworks to learn BNE across various auction mechanisms (Huang et al., 2024) via transformer architectures (Vaswani et al., 2017).

**Learning BNE in Continuous Auction Games**: Our focus aligns with the continuous value setting, where solving BNE requires optimizing each bidder's bidding function, similar to optimizing differentiable games. Despite the success of gradient-based learning in differentiable games (Daskalakis et al., 2018; Letcher et al., 2019), gradient methods face a fundamental challenge of gradient computation in auction games, due to the non-smooth nature of utility function (Bichler et al., 2021). To address this issue, existing works have employed various gradient estimation methods through evolutionary strategies (ES) (Bichler et al., 2021; Li & Wellman, 2021; Bichler et al., 2023b), or approximating the utility function to allow for smooth gradient computation (Kohring et al., 2023). Theoretically, Bichler et al. (2023c) provided convergence analysis on gradient-based methods in symmetric auction games by formulating the BNE as a variational inequality (VI) problem. Our work follows this gradient-based learning paradigm in continuous auction games, with novel gradient estimation techniques and learning objectives to improve convergence rates.

# B. Limitations and Future Directions

**Theoretical Limitations**: Our theoretical framework primarily focuses on the symmetric auction with a uniform prior following Feng et al. (2021), and employs a shared linear bidding strategy. Future research could extend these insights to more generalized settings that include a broader array of auction types and bidding strategies. Additionally, the theoretical results presented are based on exact gradients in their population form and do not incorporate empirical loss under different estimation scenarios, *e.g.,* the KDE or histogram methods when estimating the pdf of the market price distribution. Future studies could address this limitation by deriving solutions using stochastic optimization approaches, which may offer more robust and practical insights.

**Methodological Constraints**: Our work introduces a new gradient estimation method that utilizes the analytic gradient and proposes a new optimization approach, approximate BR gradient. However, the closed-form solution required by our gradient estimation method might not generalize across all possible scenarios, particularly in more complex auction settings where more sophisticated gradient estimation techniques might be necessary. Although our optimization objective operates independently of specific gradient estimation methods, its reliance on second-order gradients introduces additional complexity, necessitating a differentiable estimated distribution.

## C. Proofs

### C.1. Gradient Estimation

Following previous work (Huang et al., 2024; Bichler et al., 2023c), the *ex-post* utility $u_i$ can be reformulated as the function $\tilde{u}_i$ of competition with market price $m_i = \max_{j \neq i} b_j$:

$$u_i(v_i, b_i, b_{-i}) = \tilde{u}_i(v_i, b_i, m_i) \quad (m_i := \max_{j \neq i} b_j)$$
$$= \tilde{u}_i^+(v_i, b_i, m_i) \cdot \mathbb{I}(b_i > m_i), \tag{18}$$

where the $\tilde{u}_i^+(v_i, b_i, m_i)$ is the utility when bidder $i$ wins. We present the results in FP and SP, where $\tilde{u}_i^+(v_i, b_i, m_i) = v_i - b_i$ in FP and $\tilde{u}_i^+(v_i, b_i, m_i) = v_i - m_i$ in SP respectively. Without loss of generality, we assume:

**Assumption C.1.** The market price distribution is atomless.

The ex-interim utility's gradient can be directly computed:

$$\frac{\partial}{\partial b_i} \bar{u}_i(v_i, b_i, \beta_{-i})$$
$$= \frac{\partial}{\partial b_i} \mathbb{E}_{v_{-i}|v_i} [u_i(v_i, b_i, b_{-i}(v_{-i}))]$$
$$= \frac{\partial}{\partial b_i} \mathbb{E}_{m_i|v_i} [\tilde{u}_i^+(v_i, b_i, m_i) \cdot \mathbb{I}(b_i > m_i)]$$
$$= \frac{\partial}{\partial b_i} \int \tilde{u}_i^+(v_i, b_i, x) \cdot \mathbb{I}(b_i > x) dF_{m_i}(x)$$
$$= \frac{\partial}{\partial b_i} \int_0^{b_i} \tilde{u}_i^+(v_i, b_i, x) \cdot f_{m_i}(x) dx$$
$$= \tilde{u}_i^+(v_i, b_i, b_i) \cdot f_{m_i}(b_i) + \int_0^{b_i} \frac{\partial}{\partial b_i} \tilde{u}_i^+(v_i, b_i, x) \cdot f_{m_i}(x) dx.$$

In the case of first-price (FP) and second-price (SP) auctions, the gradient is calculated by:

$$\frac{\partial}{\partial b_i} \bar{u}_i(v_i, b_i, \beta_{-i}) = \begin{cases} (v_i - b_i) f_{m_i}(b_i) - F_{m_i}(b_i), & \text{FP} \\ (v_i - b_i) f_{m_i}(b_i), & \text{SP} \end{cases}$$

Moreover, the second-order gradient, which is used in the approximate BR method, is calculated by:

$$\frac{\partial^2}{\partial b_i^2} \bar{u}_i(v_i, b_i, \beta_{-i}) = \begin{cases} (v_i - b_i) f'_{m_i}(b_i) - 2 f_{m_i}(b_i), & \text{FP} \\ (v_i - b_i) f'_{m_i}(b_i) - f_{m_i}(b_i), & \text{SP} \end{cases}$$

Notably, this gradient formula can be extended to other settings, such as risk aversion and reserve price. For example, consider a risk-averse utility (Krishna, 2010) such that the *ex-post* utility is extended as:

$$u_i^\rho(v_i, b_i, b_{-i}) = \begin{cases} u_i(v_i, b_i, b_{-i})^\rho, & u_i \geq 0 \\ u_i(v_i, b_i, b_{-i}), & \text{else} \end{cases} \tag{19}$$

where $u_i(v_i, b_i, b_{-i})$ is the original risk-neutral utility function defined in (2) and $\rho \in (0, 1)$ is the risk-aversion weight. Take the first price auction as an example, the gradient of *ex-interim* utility in this risk-aversion setting can be computed as (denote $\bar{u}_i^\rho(v_i, b_i, \beta_{-i})$ with $\bar{u}_i^\rho$ for simplicity):

$$\frac{\partial}{\partial b_i} \bar{u}_i^\rho = \begin{cases} (v_i - b_i)^\rho f_{m_i}(b_i) - \rho(v_i - b_i)^{\rho-1} F_{m_i}(b_i), & b_i \leq v_i \\ (v_i - b_i) f_{m_i}(b_i) - F_{m_i}(b_i), & b_i > v_i \end{cases}$$

$$\frac{\partial^2}{\partial b_i^2} \bar{u}_i^\rho = \begin{cases} (v_i - b_i)^\rho f'_{m_i}(b_i) - 2\rho(v_i - b_i)^{\rho-1} f_{m_i}(b_i) \\ \quad + \rho(\rho - 1)(v_i - b_i)^{\rho-2} F_{m_i}(b_i), & b_i \leq v_i \\ (v_i - b_i) f'_{m_i}(b_i) - 2 f_{m_i}(b_i), & b_i > v_i \end{cases}$$

As for scenarios with a reserve price $r$, the ex-iterim utility is modified as:

$$\bar{u}_i(v_i, b_i, \beta_{-i}) = \begin{cases} \mathbb{E}_{v_{-i}}[(v_i - b_i) \cdot \mathbb{I}(b_i > \max\{\beta_{-i}(v_{-i}), r\})], & \text{FP}, \\ \mathbb{E}_{v_{-i}}[(v_i - \max\{\beta_{-i}(v_{-i}), r\}) \cdot \mathbb{I}(b_i > \max\{\beta_{-i}(v_{-i}), r\})], & \text{SP}. \end{cases}$$

To estimate the gradients, we can replace the original market price $m_i = \max_{j \neq i} b_j$ with a reserved version: $m_i^r = \max\{m_i, r\}$. We can estimate the distribution of the reserved market price $m_i^r$ by sampling bids $b_{-i}$, and compute the pdf $f_{m_i^r}$ and cdf $F_{m_i^r}$. Then the gradient with the reserve price can be estimated via Equation (11) by changing the distribution from $m_i$ to $m_i^r$.

## C.2. Convergence Analysis on Utility Gradient

### C.2.1. CONVERGENCE ABILITY

Here we provide the convergence ability of different MC estimation methods under the symmetric uniform first-price auction setting of Assumption 3.1. We will prove that projected gradient ascent via our derived closed-form gradient (MC on distribution) converges to BNE while the MC on gradient converges to zero-bidding:

*Proof of Proposition 3.2.* The gradient of $\bar{u}_i$ is calculated as:

$$\frac{\partial}{\partial b_i} \bar{u}_i(v_i, b_i, \beta_{-i}) = (v_i - b_i) f_{m_i}(b_i) - F_{m_i}(b_i).$$

Assuming a linear bidding function $\beta_i(v_i) = wv_i + c$, the market price distribution becomes:

$$F_{m_i}(x) = \left(\frac{x - c}{w}\right)^{n-1}, f_{m_i}(x) = \frac{n-1}{w}\left(\frac{x - c}{w}\right)^{n-2},$$

thus, the gradient of $\bar{u}_i$ w.r.t. $b_i = \beta_i(v_i)$ is:

$$\frac{\partial}{\partial b_i} \bar{u}_i(v_i, \beta_i(v_i), \beta_{-i}) = \frac{n - 1 - nw}{w} v_i^{n-1} - c \frac{n-1}{w} v_i^{n-2}.$$

By applying the chain rule, the derivatives of $\theta_i = (w, c)^\intercal$ are

$$\nabla_w \bar{u}_i(v_i, \beta_i(v_i), \beta_{-i}) = v_i \cdot \frac{\partial}{\partial b_i} \bar{u}_i(v_i, \beta_i(v_i), \beta_{-i}),$$

$$\nabla_c \bar{u}_i(v_i, \beta_i(v_i), \beta_{-i}) = \frac{\partial}{\partial b_i} \bar{u}_i(v_i, \beta_i(v_i), \beta_{-i}).$$

Given that $\nabla_{\theta_i} U_i(\beta_{\theta_i}, \beta_{-i}) = \mathbb{E}_{v_i} \nabla_{\theta_i} \bar{u}_i(v_i, \beta_{\theta_i}(v_i), \beta_{-i})$, we have:

$$\nabla_w U_i(\beta_i, \beta_{-i}) = \frac{n - 1 - nw}{(n+1)w} - \frac{(n-1)c}{nw}$$

$$\nabla_c U_i(\beta_i, \beta_{-i}) = \frac{n - 1 - nw}{nw} - \frac{c}{w}.$$

Since the parameters are constrained to be non-negative, we utilize projected gradient ascent:

$$\theta_i^{t+1} = \text{Proj}(\theta_i^t + \alpha \nabla_{\theta_i} U_i),$$

where the projection operator is defined as:

$$\text{Proj}(\theta_i) = \text{Proj}(w, c) = \min_{w', c' \geq 0} \|w' - w\|_2^2 + \|c' - c\|_2^2$$

$$= (\max\{w, 0\}, \max\{c, 0\})^\intercal.$$

To prove that projected gradient ascent converges to the BNE strategy $\theta_i^* = (w^*, c^*)^\intercal = (\frac{n-1}{n}, 0)^\intercal$, we seek to show that:

$$\langle \nabla_{\theta_i} U_i, \theta_i^* - \theta_i \rangle \geq 0.$$

This condition is sufficient for convergence since:

$$
\begin{aligned}
\|\theta_i^{t+1} - \theta_i^*\|^2 &= \|\theta_i^{t+1} - \mathrm{Proj}(\theta_i^*)\|^2 \\
&= \|\mathrm{Proj}(\theta_i^t + \alpha \nabla_{\theta_i^t} U_i) - \mathrm{Proj}(\theta_i^*)\|^2 \\
&\leq \|\theta_i^t + \alpha \nabla_{\theta_i^t} U_i - \theta_i^*\|^2 \quad \text{(non-expansion)} \\
&= \|\theta_i^t - \theta_i^*\|^2 - 2\alpha \langle \nabla_{\theta_i^t} U_i, \theta_i^* - \theta_i^t \rangle + \alpha^2 \|\nabla_{\theta_i^t} U_i\|^2 \\
&\leq \|\theta_i^t - \theta_i^*\|^2 \quad \text{(when } \alpha \text{ is small)}
\end{aligned}
$$

Denoting $\delta_w = w - \frac{n-1}{n}$, then $n - 1 - nw = -n\delta_w$, and the gradient can be written as:

$$
\nabla_{\theta_i} U_i = \begin{pmatrix} \nabla_w U_i \\ \nabla_c U_i \end{pmatrix} = \begin{pmatrix} \frac{-n\delta_w}{(n+1)w} - \frac{(n-1)c}{nw} \\ \frac{-\delta_w}{w} - \frac{c}{w} \end{pmatrix}, \theta_i^* - \theta_i = \begin{pmatrix} -\delta_w \\ -c \end{pmatrix}.
$$

Considering the different cases for $\delta_w$:

- When $\delta_w = 0$: The dot-product equals $c^2/w \geq 0$ and the equality only occurs at $c = 0$, *i.e.*, $\theta_i = \theta_i^*$.

- When $\delta_w \neq 0$: The dot-product is:

$$
-\delta_w \nabla_w U_i - c \nabla_c U_i = \frac{1}{w} \left( \frac{n}{n+1} \delta_w^2 + \frac{2n-1}{n} c\delta_w + c^2 \right).
$$

Given that the discriminator (w.r.t. c) $\Delta = \delta_w^2 \frac{1-3n}{n^2(n+1)} < 0$, the dot-product remains positive.

Thus, the gradient satisfies $\langle \nabla_{\theta_i} U_i, \theta_i^* - \theta_i \rangle \geq 0$, where the equality holds only when $\theta_i = \theta_i^*$, ensuring convergence to $\theta_i^*$. We've also provided a gradient flow of $\nabla_{\theta_i} U_i$ in Fig. 6, from which it can be observed that the gradient vector always has a positive part pointing to the BNE point.

For MC estimation on gradient, the parameters update is given by:

$$
\begin{aligned}
\frac{\partial}{\partial b_i} \bar{u}_i(v_i, b_i, \beta_{-i}) &= -\Pr(i \text{ wins}) = -F_{m_i}(b_i), \\
\nabla_w \bar{u}_i(v_i, \beta_i(v_i), \beta_{-i}) &= -v_i^n, \\
\nabla_c \bar{u}_i(v_i, \beta_i(v_i), \beta_{-i}) &= -v_i^{n-1},
\end{aligned}
$$

thus:

$$
\begin{aligned}
\nabla_w U_i(\beta_i, \beta_{-i}) &= -\frac{1}{n+1} \\
\nabla_c U_i(\beta_i, \beta_{-i}) &= -\frac{1}{n}.
\end{aligned}
$$

The gradient is always negative, causing the parameters to converge to $w, c = 0$ with the projection operator. This results in the zero-bidding strategy $\beta_i(v_i) = 0$. $\square$

We've also validated this phenomenon experimentally in the symmetric first-price auction setting with $n = 2$ bidders who share a 3-layer MLP network as their bidding function and employ stochastic gradient descent (SGD) to optimize the network. As illustrated in Fig. 7, the network employing our gradient estimation method in Eq. (11) successfully converges to the BNE. In contrast, the network trained using the MC estimation on gradient in Eq. (10) converges toward zero bidding.

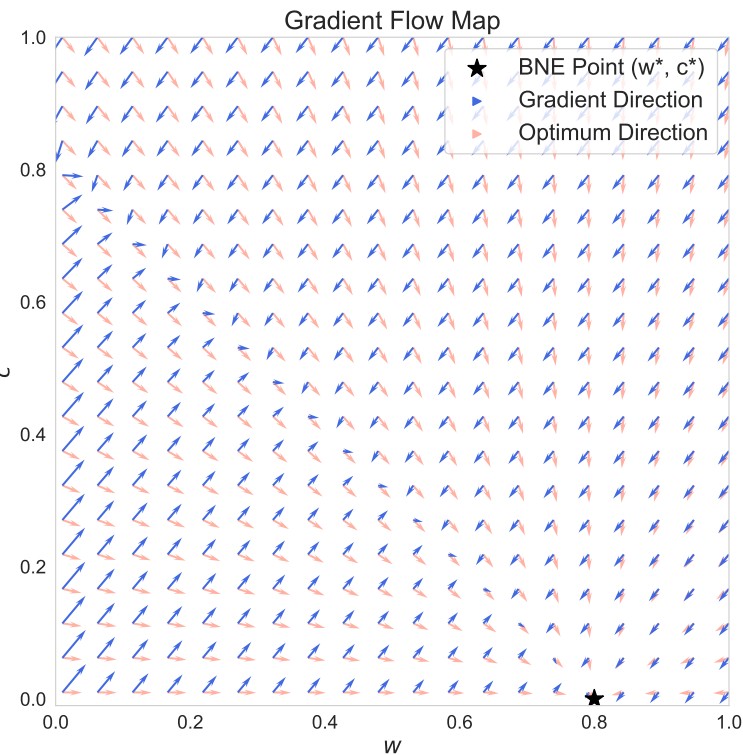

*Figure 6.* Gradient flow of $\nabla_{\theta_i} U_i$ when $n = 5$

### C.2.2. CONVERGENCE RATE

**Upper bound.** Here we prove the $\Theta(n^2)$ condition number of $U_i$'s Hessian matrix (Lemma 4.1) and $\mathcal{O}(n^2 \log \frac{1}{d})$ local convergence rate of projected gradient ascent under the symmetric uniform first-price auction setting of Assumption 3.1:

*Proof of Lemma. 4.1 and Theorem. 4.2 (Part I: upper bound).* Firstly we examine the Hessian matrix of $U_i$:

$$\nabla_{\theta_i}^2 U_i = \mathbb{E}_{v_i} \left[ \frac{\partial^2}{\partial b_i^2} \bar{u}_i \nabla_{\theta_i} \bar{\beta}_i(v_i) \nabla_{\theta_i} \bar{\beta}_i(v_i)^{\mathsf{T}} + \frac{\partial}{\partial b_i} \bar{u}_i \nabla_{\theta_i}^2 \bar{\beta}_i(v_i) \right].$$

With the second gradient of $\frac{\partial^2}{\partial b_i^2} \bar{u}_i(v_i, b_i, \beta_{-i})$ at $b_i = \beta_{\theta_i}(v_i)$ being (assuming $n > 2$):

$$\frac{\partial^2}{\partial b_i^2} \bar{u}_i = \frac{(n-1)(n-2-nw)}{w^2} v_i^{n-2} - c \frac{(n-1)(n-2)}{w^2} v_i^{n-3},$$

the Hessian matrix is expressed as (Let $S_w = nw + 2 - n$):

$$\boldsymbol{H}(w, c) = - \begin{bmatrix} \frac{(n-1)S_w}{(n+1)w^2} + \frac{c(n-1)(n-2)}{nw^2} & \frac{(n-1)S_w}{nw^2} + \frac{c(n-2)}{w^2} \\ \frac{(n-1)S_w}{nw^2} + \frac{c(n-2)}{w^2} & \frac{S_w}{w^2} + \frac{c(n-1)}{w^2} \end{bmatrix}.$$

For the local convergence rate, we define a neighbor region of the BNE point:

$$\mathcal{N}(w^*, c^*) := \{(w, c)^{\mathsf{T}} \in R_+^2 | |w - w^*|^2 + |c - c^*|^2 \leq r^2\},$$

where we set a $\Theta(\frac{1}{n})$ radius: $r = \frac{1}{2n}$, so that $S_w \in [\frac{1}{2}, \frac{3}{2}]$. Now we analyze the property of the Hessian matrix within this neighborhood, which satisfies:

- Every element of $-\boldsymbol{H}$ is positive and in the order of $\Theta(1)$, thus the max eigenvalue $\lambda_{\max} = \mathcal{O}(1)$;

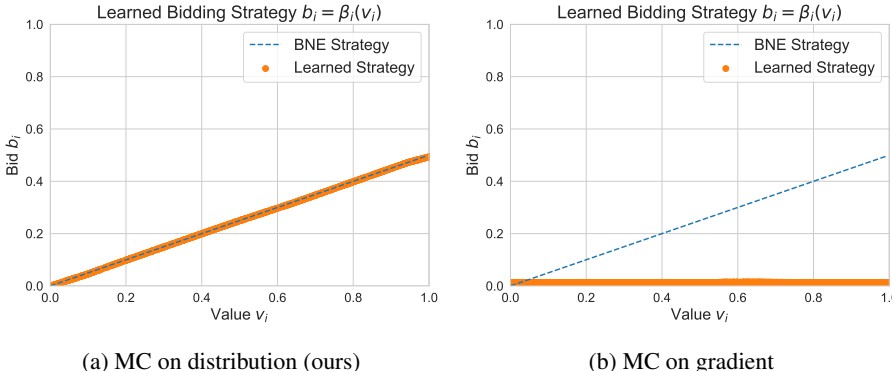

(a) MC on distribution (ours)          (b) MC on gradient

*Figure 7.* Convergence results of different gradient estimation methods. The strategy converging to BNE in (a) is learned via the proposed MC estimation on distribution, while the zero-bidding strategy in (b) is learned via the MC estimation on gradient.

- The determinant is positive:

$$\mathrm{Det}(-\boldsymbol{H}) = \frac{1}{w^4}\left[\frac{n-1}{(n+1)n^2}S_w^2 + \frac{2(n-1)}{n(n+1)}cS_w + \frac{n-2}{n}c^2\right] > 0;$$

- With the eigenvalues $\lambda$ satisfying:

$$\mathrm{Det}(-\boldsymbol{H} - \lambda I) = \lambda^2 - \left(\frac{2nS_w}{(n+1)w^2} + \frac{2(n-1)^2c}{nw^2}\right)\lambda + \mathrm{Det}(-\boldsymbol{H}),$$

one can verify that the discriminator of this quadratic (w.r.t. $\lambda$) is positive, so that $-\boldsymbol{H}$ has 2 positive real eigenvalues, $-\boldsymbol{H}$ is positive definite, and $\lambda_{\max} \geq \frac{nS_w}{(n+1)w^2} + \frac{(n-1)^2c}{nw^2} = \Omega(1)$;

- The determinant $\mathrm{Det}(-\boldsymbol{H})$ is in the order of $\Theta(\frac{1}{n^2})$, the max eigenvalue $\lambda_{\max}$ is in the order of $\Theta(1)$ so that the condition number $\kappa = \frac{\lambda_{\max}}{\lambda_{\min}} = \frac{\lambda_{\max}^2}{\mathrm{Det}(-\boldsymbol{H})}$ is $\boxed{\Theta(n^2)}$.

Having examined the local property of $\boldsymbol{H}(w^*, c^*)$, we can now prove the local convergence rate within $\mathcal{N}(w^*, c^*)$. Since we've established in previous proof that:

$$\|\theta_i^{t+1} - \theta_i^*\|^2 \leq \|\theta_i^t - \theta_i^*\|^2,$$

the gradient update will keep the parameter inside the neighbor region $\mathcal{N}(w^*, c^*)$, making the negative utility function always convex and the condition number $\kappa$ always being $\Theta(n^2)$. Thus we can directly apply the results of general convex optimization problems: as established by Bubeck (2015) (Theorem 3.12), with stepsize $\alpha = \frac{2}{\lambda_{\max} + \lambda_{\min}}$, the projected gradient ascent update $\theta_i^{t+1} = \mathrm{Proj}(\theta_i^t + \alpha\nabla_{\theta_i}U_i)$ satisfies:

$$\|\theta_i^t - \theta_i^*\| \leq \left(1 - \frac{2}{\kappa + 1}\right)\|\theta_i^t - \theta_i^*\|.$$

With $\kappa$ being $\Theta(n^2)$, it will take $\boxed{\mathcal{O}(n^2\log\frac{1}{d})}$ steps for $\|\theta_i^t - \theta_i^*\| \leq d$. □

**Lower Bound.** Here we prove the convergence rate's lower bound of the utility gradient method.

*Proof of Theorem 4.2 (Part II: lower bound).* As previously stated, let $\delta_w = w - \frac{n-1}{n}$, the utility gradient update (w.r.t. $\delta_w$ and $c$) can be written as:

$$\begin{pmatrix}\delta_w^{t+1} \\ c^{t+1}\end{pmatrix} = \begin{pmatrix}\delta_w^t \\ c^t\end{pmatrix} - \frac{\alpha}{\delta_w^t + \frac{n-1}{n}} \cdot \begin{pmatrix}\frac{n\delta_w^t}{n+1} + \frac{(n-1)c^t}{n} \\ \delta_w^t + c^t\end{pmatrix}.$$

We consider an initial point $\theta^0 = (\frac{n-2}{n}, \frac{1}{n+1})$ thus $\delta_w^0 = -\frac{1}{n}$. Applying the update rule to the initial point:

$$\begin{pmatrix} \delta_w^1 \\ c^1 \end{pmatrix} = \begin{pmatrix} -\frac{1}{n} \\ \frac{1}{n+1} \end{pmatrix} - \frac{\alpha n}{n-2} \cdot \begin{pmatrix} -\frac{1}{n^2+n} \\ -\frac{1}{n^2+n} \end{pmatrix},$$

to ensure convergence, we need to make sure: $(\delta_w^1)^2 + (c^1)^2 \le (\delta_w^0)^2 + (c^0)^2$, resulting in $\alpha < \frac{n-2}{n}$. We first claim that during optimization, $\delta_w^t \in [-\frac{1}{n}, 0]$ and $c^t \ge 0$ (so that projected gradient ascent degrades to the vanilla gradient ascent), which will be verified by induction later. Then we have:

$$\frac{\alpha}{\delta_w^t + \frac{n-1}{n}} \in [0, 1].$$

For convergence analysis, we separate the update rule into two independently updating terms, denoted as $a^t = \delta_w^t + k_1 c^t$ and $b^t = \delta_w^t + k_2 c^t$, where $k_1, k_2$ are two roots of the following quadratics:

$$-\frac{nkc}{n+1} + \frac{(n-1)c}{n} + k(-kc + c) = 0 \iff k\left(\frac{n}{n+1} + k\right) = \frac{n-1}{n} + k.$$

This quadratics is derived by letting $\nabla_{\delta_w} U = -k \nabla_c U$ and replace $\delta_w = -kc$. Without loss of generality, let $k_1 < k_2$, one may verify that:

$$-\frac{n}{n+1} < k_1 < -\frac{n}{n+1} + \frac{1}{n^2}, \quad \frac{n^2-1}{n^2} < k_2 < 1.$$

With the equally scaled gradient and parameter $\nabla_{\delta_w} U / \nabla_c U = \delta_w / c$, we have:

$$\delta_w^{t+1} + kc^{t+1} = \delta_w^t + kc^t - \frac{\alpha}{\delta_w^t + \frac{n-1}{n}}\left((\frac{n}{n+1} + k)\delta_w^t + (\frac{n-1}{n} + k)c^t\right) = \left(1 - \frac{\alpha}{\delta_w^t + \frac{n-1}{n}} \cdot (\frac{n}{n+1} + k)\right)(\delta_w^t + kc^t),$$

which means $a^{t+1} = \left(1 - \frac{\alpha}{\delta_w^t + \frac{n-1}{n}} \cdot (\frac{n}{n+1} + k_1)\right) a^t$ and $b^{t+1} = \left(1 - \frac{\alpha}{\delta_w^t + \frac{n-1}{n}} \cdot (\frac{n}{n+1} + k_2)\right) b^t$. Since the optimum of $a$ and $b$ are both $a^* = b^* = 0$, the convergence of $a^t$ and $b^t$ can be directly examined via the multiplier $1 - \frac{\alpha}{\delta_w^t + \frac{n-1}{n}} \cdot (\frac{n}{n+1} + k)$. Given $\frac{\alpha}{\delta_w^t + \frac{n-1}{n}} < 1$, the convergence rates are as follows:

- For $a^t$ with $-\frac{n}{n+1} < k_1 < -\frac{n}{n+1} + \frac{1}{n^2}$, we have $(\frac{n}{n+1} + k_1) \in (0, \frac{1}{n^2})$, so $a^t$ converges to $a^* = 0$ at $\Omega(n^2 \log \frac{1}{d})$, with the lower bound obtained with some constant-order $\alpha = \Theta(1)$.

- For $b^t$ with $\frac{n^2-1}{n^2} < k_2 < 1$, we have $(\frac{n}{n+1} + k_1) \in (2 - \frac{1}{n+1} - \frac{1}{n^2}, 2 - \frac{1}{n+1})$, so $b^t$ converges to $b^0 = 0$ at $\mathcal{O}(n \log \frac{1}{d})$ with the constant-order $\alpha$.

For large $n$, the initial $a^0 = -\frac{1}{n} + \frac{k_1}{n+1} \approx -\frac{2}{n}$ while $b^0 = -\frac{1}{n} + \frac{k_2}{n+1} \approx -\frac{1}{n^2}$, so $a^0 \gg b^0$. This imbalance further expands due to then $\Omega(n^2 \log \frac{1}{d})$ convergence rate for $a^t$ while $\mathcal{O}(n \log \frac{1}{d})$ rate for $b^t$. As a result, the distance to optimum is dominated by $a^t$:

$$\begin{aligned} \|\theta^t - \theta^*\|^2 = (\delta_w^t)^2 + (c^t)^2 &= \left(\frac{k_2 a^t - k_1 b^t}{k_2 - k_1}\right)^2 + \left(\frac{a^t - b^t}{k_1 - k_2}\right)^2 \\ &= \frac{1}{(k_1 - k_2)^2}[(k_2^2 + 1)(a^t)^2 - 2(k_1 k_2 + 1)a^t b^t + (k_1^2 + 1)(b^t)^2] \\ &\approx \frac{k_2^2 + 1}{(k_1 - k_2)^2}(a^t)^2. \end{aligned}$$

Resulting in an $\boxed{\Omega(n^2 \log \frac{1}{d})}$ convergence rate for $\theta$.

Additionally, the condition $\delta_w^t \in [-\frac{1}{n}, 0]$ can be verified via induction (we consider a large $n$ for simplicity):

- Initially, we have $\delta_w^0 = -\frac{1}{n}$, $a^0 \approx -\frac{2}{n}$ and $b^0 \approx -\frac{1}{n^2}$.

- With $a^{t+1} = \left(1 - \frac{\alpha}{\delta_w^t + \frac{n-1}{n}} \cdot \left(\frac{n}{n+1} + k_1\right)\right) a^t$, the multiplier of $a^t$ is positive by induction, so the sign of $a^{t+1}$ and $a^t$ are the same. And $a^{t+1} \gg b^{t+1}$ holds by the update rule. So $\delta_w^{t+1} = \frac{k_2 a^{t+1} - k_1 b^{t+1}}{k_2 - k_1} \approx \frac{k_2 a^{t+1}}{k_2 - k_1} < 0$. With $a^t$ converges to 0, the shrinking magnitudes ensures $\delta_w^{t+1}$ still in $[-\frac{1}{n}, 0]$.

Similarly, the projected gradient ascent becomes the vanilla gradient ascent method since $c^t = \frac{a^t - b^t}{k_1 - k_2} \geq 0$. □

We provide illustrative examples for the constructed lower bound. Given the initial point $\theta^0 = \left(\frac{n-2}{n}, \frac{1}{n+1}\right)$, we set the learning rate $\alpha$ to $\frac{n-2}{n}$, and the optimization trajectory in shown in Fig. 8. As predicted, the point quickly converges to around the line $\delta_w + k_2 c = 0$ (*i.e.*, $b^t = 0$) but takes much more steps to converge to the optimum due to the slow convergence of $a^t$.

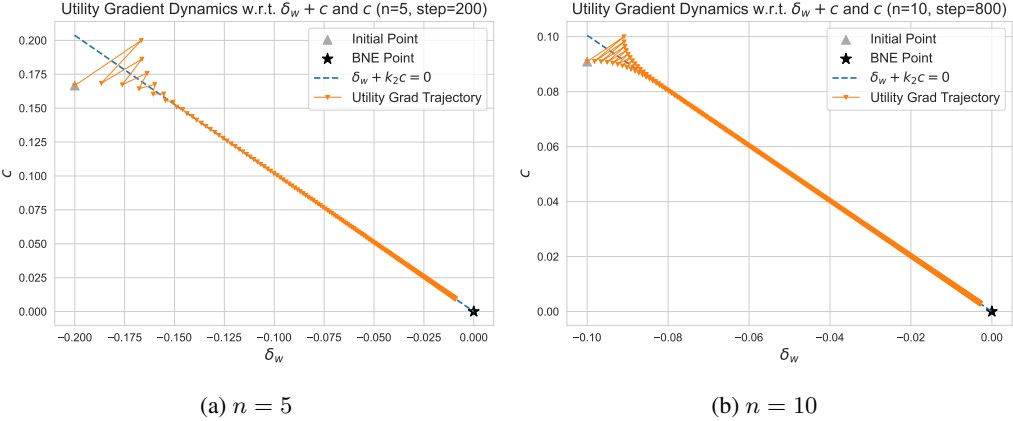

(a) $n = 5$            (b) $n = 10$

*Figure 8.* An illustration of our construction of utility gradient's lower bound. The figure illustrates the parameter quickly converges to the vicinity of the line $\delta_w + k_2 c = 0$, but slowly converges to optimum. We use 200 steps for the $n = 5$ case and employ $200 \times 2^2 = 800$ iterations for the $n = 5 \times 2 = 10$ case to show the quadratically growing complexity as $n$ increases.

### C.2.3. ACCELERATED METHODS

One may argue that the choice of projected gradient ascent rather than some accelerated gradient method, such as the Momentum or Nesterov method, might harm the convergence rate. However, it can be verified that even with Nesterov (or FISTA algorithm in the constrained optimization setting), which is known to achieve the lower bound time complexity of first-order algorithm (Nesterov, 2018), the local convergence rate of the utility gradient is $\mathcal{O}(n \log \frac{1}{d})$. This convergence rate can be directly derived via Theorem 3.15 from (Bubeck, 2015) like the previous results, so we omit it here.

We've also included a comparison with the Nesterov method in Fig. 9. As depicted in the figure, using fixed steps, the accelerated Nesterov method seems to alleviate the problem of slow convergence when $n = 10$, but it still cannot converge if the bidder number $n$ increases further. In contrast, our proposed BR gradient methods successfully converge to the BNE point in both cases.

### C.3. Convergence Analysis on BR Gradient

#### C.3.1. THE BR DISTANCE OBJECTIVE

Here we prove that the Best Response Distance objective is an upper bound of the approximation factor $\epsilon$ in $\epsilon$-BNE, as stated in Lemma 4.3.

*Proof of Lemma 4.3.* Let the approximation factor $\epsilon$ being:

$$\epsilon = \max_i \mathbb{E}_{v_i \sim \mathcal{F}_i}\left[\bar{u}_i(v_i, \mathrm{br}_i(v_i, \hat{\beta}_{-i}), \hat{\beta}_{-i})) - \bar{u}_i(v_i, \hat{\beta}_i(v_i), \hat{\beta}_{-i})\right].$$

then $\hat{\beta}$ is an $\epsilon$-BNE by definition. For the BR distance, let $\bar{u}_i(b_i) = \bar{u}_i(v_i, b_i, \hat{\beta}_{-i})$ and $\mathrm{br}_i = \mathrm{br}_i(v_i, \hat{\beta}_{-i})$ for short, we

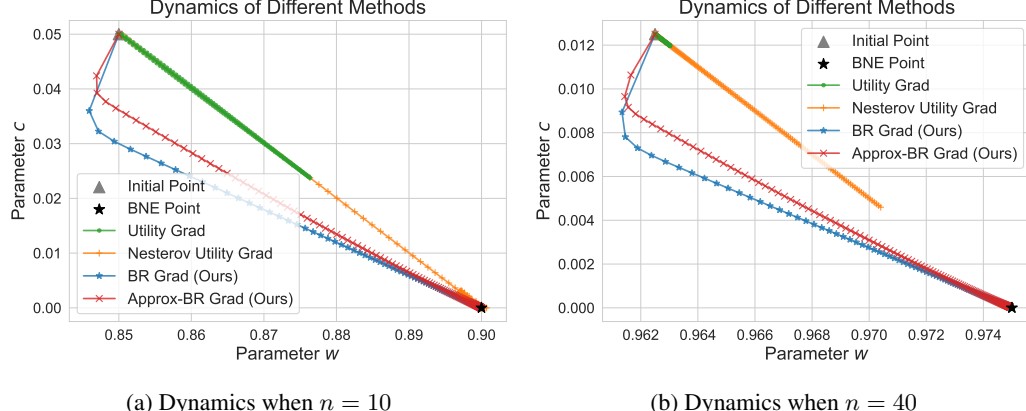

(a) Dynamics when $n = 10$         (b) Dynamics when $n = 40$

*Figure 9.* Learning dynamics of different methods in the setting of Proposition 3.2. The initial point $\theta_i^{t=0}$ is picked at the vicinity of $\mathcal{N}(\theta^*)$, such that $\theta_i^{t=0} = \theta_i^* + (\frac{-1}{2n}, \frac{1}{2n})^\intercal$. All methods run 150 steps by default and we plot the best-performing results after learning-rate grid search.

have:

$$\mathcal{L}_{\mathrm{BR}}(\hat{\beta}) = \sum_{i \leq n} \frac{1}{2} \mathbb{E}_{v_i} \left[ \|\hat{\beta}_i(v_i) - \mathrm{br}_i\|^2 \right]$$

$$\geq \max_i \frac{1}{2} \mathbb{E}_{v_i} \left[ \|\hat{\beta}_i(v_i) - \mathrm{br}_i\|^2 \right]$$

$$\geq \max_i \frac{1}{2} \left( \mathbb{E}_{v_i} [\|\hat{\beta}_i(v_i) - \mathrm{br}_i\|] \right)^2 \quad \text{(Jensen's inequality)}$$

$$\geq \max_i \frac{1}{2} \left( \mathbb{E}_{v_i} [\frac{1}{L} |\bar{u}_i(\hat{\beta}_i(v_i)) - \bar{u}_i(\mathrm{br}_i)|] \right)^2 \quad \text{($L$-Lipschitz)}$$

$$= \frac{1}{2L^2} \epsilon^2 .$$

Thus $\epsilon$ is bounded by $\sqrt{2\mathcal{L}_{\mathrm{BR}}(\hat{\beta})} \cdot L$.      □

### C.3.2. CONVERGENCE RATE

Here we prove the $\mathcal{O}(\log \frac{1}{d})$ convergence rate of the projected gradient descent via the BR gradient (*i.e.,* optimizing the BR distance objective), under the symmetric uniform first-price auction setting of Assumption 3.1:

*Proof of Theorem 4.4.* The BR gradient is defined as:

$$g_{\mathrm{br}_i} = \mathbb{E}_{v_i} \left[ (wv_i + c - \mathrm{br}_i(v_i)) \nabla_{\theta_i} \beta_{\theta_i}(v_i) \right]$$
$$\text{where } \mathrm{br}_i(v_i) = \arg\max_{b_i} \bar{u}_i(v_i, b_i, \beta_{-i}).$$

Let $\partial \bar{u}_i / \partial b_i = 0$, the best-response can be calculated as:

$$\mathrm{br}_i(v_i) = \frac{n-1}{n} v_i + \frac{c}{n},$$

therefore the BR gradient can be calculated as:

$$g_{\mathrm{br}_i} = \begin{pmatrix} \frac{1}{3}(w - \frac{n-1}{n}) + \frac{n-1}{2n}c \\ \frac{1}{2}(w - \frac{n-1}{n}) + \frac{n-1}{n}c \end{pmatrix}.$$

So the update rule of the BR gradient can be written as:

$$\theta_i^{t+1} = \text{Proj}\left(\theta_i^t - \alpha g_{\text{br}_i}^t\right),$$

$$\text{where } \theta_i^t - \alpha g_{\text{br}_i}^t = \begin{pmatrix} w^t - \frac{\alpha}{3}\delta_w^t - \frac{(n-1)\alpha}{2n}c^t \\ c^t - \frac{\alpha(n-1)}{n}c^t - \frac{\alpha}{2}\delta_w^t \end{pmatrix}.$$

To measure the distance to $\theta^*$, let:

$$\theta_i^t - \alpha g_{\text{br}_i}^t - \theta_i^* = \begin{pmatrix} (1 - \frac{\alpha}{3})\delta_w^t - \frac{(n-1)\alpha}{2n}c^t \\ (1 - \frac{\alpha(n-1)}{n})c^t - \frac{\alpha}{2}\delta_w^t \end{pmatrix} = A(\theta_i^t - \theta_i^*)$$

where the matrix $A$ is defined as:

$$A = \begin{bmatrix} 1 - \frac{\alpha}{3} & -\frac{(n-1)\alpha}{2n} \\ -\frac{\alpha}{2} & 1 - \frac{\alpha(n-1)}{n} \end{bmatrix}.$$

So the distance to $\theta_i^*$ satisfies:

$$\begin{aligned}
\|\theta_i^{t+1} - \theta_i^*\| &= \|\text{Proj}\left(\theta_i^t - \alpha g_{\text{br}_i}^t\right) - \text{Proj}(\theta_i^*)\| \\
&\leq \|\theta_i^t - \alpha g_{\text{br}_i}^t - \theta_i^*\| \\
&= \|A(\theta_i^t - \theta_i^*)\| \\
&\leq \|A\| \cdot \|\theta_i^t - \theta_i^*\|.
\end{aligned}$$

So the problem converts to examine the spectral radius of $A$. Let the stepsize being $\alpha = \frac{1}{2}$, we have:

$$A = \begin{bmatrix} \frac{5}{6} & -\frac{(n-1)}{4n} \\ -\frac{1}{4} & \frac{n+1}{2n} \end{bmatrix}.$$

By simple calculation, one can verify that:

1. The matrix $A$ is positive-definite;

2. The spectral radius of $A$ (*i.e.,* max eigenvalue in the positive definite case) is smaller than $\frac{39}{40}$.

Thus we have:

$$\begin{aligned}
\|\theta_i^t - \theta_i^*\| &\leq \|A\| \cdot \|\theta_i^{t-1} - \theta_i^*\| \\
&\leq \frac{39}{40}\|\theta_i^{t-1} - \theta_i^*\| \\
&\leq (\frac{39}{40})^t\|\theta_i^0 - \theta_i^*\|.
\end{aligned}$$

Thus it will take $\boxed{\mathcal{O}(\log\frac{1}{d})}$ steps for $\|\theta_i^t - \theta_i^*\| \leq d$. $\qquad\square$

### C.4. Convergence Analysis on Approximate BR Gradient

Here we prove the local convergence results of the approximate BR gradient under Assumption 3.1:

*Proof of Theorem 4.5.* In a small neighbor region of $\theta_i^*$, such as the previously defined $\mathcal{N}(w^*, c^*)$, the *ex-interim* utility function is guaranteed to be concave, thus the approximate BR gradient is computed via the second-order approximation (assuming $n > 2$):

$$\begin{aligned}
\widehat{\text{br}}_i(v_i, \beta_{-i}) &= \beta_i(v_i) - \frac{\frac{\partial}{\partial b_i}\bar{u}_i(v_i, b_i, \beta_{-i})}{\frac{\partial^2}{\partial b_i^2}\bar{u}_i(v_i, b_i, \beta_{-i})} \\
&= \beta_i(v_i) - \frac{wv_i[(n-1-nw)v_i - c(n-1)]}{(n-1)[(n-2-nw)v_i - c(n-2)]}.
\end{aligned}$$

The gradient dynamics satisfies:

$$\begin{pmatrix} w^{t+1} \\ c^{t+1} \end{pmatrix} = \begin{pmatrix} w^t - \alpha \mathbb{E}_{v_i}[v_i(\beta_i(v_i) - \widehat{\mathrm{br}}_i] \\ c^t - \alpha \mathbb{E}_{v_i}[\beta_i(v_i) - \widehat{\mathrm{br}}_i] \end{pmatrix}.$$

Assume the gradients can be locally approximated as a linear function of $\theta_i^t - \theta_i^*$ (which we will validate later), *i.e.*,

$$\begin{pmatrix} \mathbb{E}_{v_i}[v_i(\beta_i(v_i) - \widehat{\mathrm{br}}_i] \\ \mathbb{E}_{v_i}[\beta_i(v_i) - \widehat{\mathrm{br}}_i] \end{pmatrix} = A(\theta_i^t - \theta_i^*).$$

Then the gradient dynamics be reformulated as:

$$\theta_i^{t+1} - \theta_i^* = (I - \alpha A) \cdot (\theta_i^t - \theta_i^*).$$

Similarly, we need to examine the spectral radius of $I - \alpha A$. Consider the convex case, *i.e.*, $A$ is positive-definite so that $\lambda_{\min}(A) > 0$, to make sure the spectral radius $\|I - \alpha A\|$ less than 1, we let $0 < \alpha < 2/\lambda_{\max}(A)$. Then the minimal spectral radius is:

$$\min_{\alpha \in (0, \frac{2}{\lambda_{\max}(A)})} \{\max(|1 - \alpha\lambda_{\max}(A)|, |1 - \alpha\lambda_{\min}(A)|)\},$$

which is satisfied when $\alpha^* = \frac{2}{\lambda_{\min}(A) + \lambda_{\max}(A)}$. (Note that this is the learning rate in the previous utility gradient's proof.) With this learning rate, we have (denote the condition number as $\kappa(A) = \lambda_{\max}(A)/\lambda_{\min}(A)$):

$$\begin{aligned} \|\theta_i^t - \theta_i^*\| &\leq \|I - \alpha A\| \cdot \|\theta_i^{t-1} - \theta_i^*\| \\ &\leq \|I - \alpha^* A\| \cdot \|\theta_i^{t-1} - \theta_i^*\| \\ &= \frac{\lambda_{\max} - \lambda_{\min}}{\lambda_{\max} + \lambda_{\min}} \cdot \|\theta_i^{t-1} - \theta_i^*\| \\ &= \frac{\kappa - 1}{\kappa + 1} \cdot \|\theta_i^{t-1} - \theta_i^*\| \\ &\leq \exp\left(-\frac{4t}{\kappa + 1}\right) \|\theta_i^0 - \theta_i^*\|. \end{aligned}$$

resulting in an $\mathcal{O}(\kappa(A) \log \frac{1}{d})$ convergence rate with optimal learning rate.

This explains the $\mathcal{O}(n^2 \log \frac{1}{d})$ local convergence rate for the utility gradient method since the gradient of the negative utility function can be locally approximated linearly by $-\boldsymbol{H} \cdot (\theta_i^t - \theta_i^*)$ and $-\boldsymbol{H}$ has an $\mathcal{O}(n^2)$ condition number.

For the approximate BR gradient method, we now derive a similar linear system for the gradients. Consider a neighbor region with radius $r$:

$$\mathcal{N}'(r) = \{\theta_i \in \mathbb{R}_+^2 | \|\theta_i - \theta_i^*\| \leq r\},$$

denote $\delta_w = w - \frac{n-1}{n}$, we have $\delta_w, c$ in the order of $\mathcal{O}(r)$.

The gradient of $c$ is:

$$\begin{aligned} & \mathbb{E}_{v_i}[\beta_i(v_i) - \widehat{\mathrm{br}}_i] \\ =& \mathbb{E}_{v_i} \frac{wv_i[(n-1-nw)v_i - c(n-1)]}{(n-1)[(n-2-nw)v_i - c(n-2)]} \\ =& \frac{w}{n-1} \mathbb{E}_{v_i} v_i \frac{n\delta_w v_i + c(n-1)}{(n\delta_w + 1)v_i + c(n-2)} \\ =& \frac{w}{n-1} \int_0^1 \frac{n\delta_w v_i + c(n-1)}{(n\delta_w + 1)v_i + c(n-2)} v_i dv_i \end{aligned}$$

Let $a = n\delta_w = \mathcal{O}(nr), b = c(n-1) = \mathcal{O}(nr), e = n\delta_w + 1 = 1 + \mathcal{O}(nr), d = c(n-2) = \mathcal{O}(nr)$. The integral can be computed as:

$$\begin{aligned} \mathrm{integral} =& \frac{ae - 2ad + 2be}{2e^2} - \frac{d(be - ad)}{e^3} \log(\frac{e+d}{d}) \\ =& \frac{a}{2e} + \frac{b}{e} + \mathcal{O}((nr)^2 \log \frac{1}{nr}). \end{aligned}$$

With some small $r$, such as $r = \mathcal{O}(\frac{1}{n^2})$, we can omit the high-order infinitesimals and get:

$$\mathbb{E}_{v_i}[\beta_i(v_i) - \widehat{\mathrm{br}}_i] \approx \frac{w}{n-1} \cdot \left(\frac{n\delta_w}{2}, c(n-1)\right)^{\mathsf{T}}$$

$$= \frac{w}{n-1} \cdot (\frac{n}{2}, n-1) \cdot \begin{pmatrix} \delta_w \\ c \end{pmatrix}.$$

Similarly, the gradient of $w$ is:

$$\mathbb{E}_{v_i} v_i[\beta_i(v_i) - \widehat{\mathrm{br}}_i]$$

$$= \frac{w}{n-1} \int_0^1 \frac{n\delta_w v_i + c(n-1)}{(n\delta_w + 1)v_i + c(n-2)} v_i^2 dv_i$$

The integral is calculated as:

$$\mathrm{integral} = \frac{1}{6e^3} \left[ a(6d^2 - 3ed + 2e^2) + 3eb(e - 2d) \right]$$

$$+ \frac{d^2}{e^4}(eb - ad) \log(\frac{d+e}{d})$$

$$= \frac{a}{3e} + \frac{b}{2e} + \mathcal{O}((nr)^2).$$

With small enough $r$ we can omit the high-order infinitesimals and get:

$$\mathbb{E}_{v_i} v_i[\beta_i(v_i) - \widehat{\mathrm{br}}_i] \approx \frac{w}{n-1} \cdot \left(\frac{n\delta_w}{3}, \frac{c(n-1)}{2}\right)^{\mathsf{T}}$$

$$= \frac{w}{n-1} \cdot (\frac{n}{3}, \frac{n-1}{2}) \cdot \begin{pmatrix} \delta_w \\ c \end{pmatrix}$$

Combining the 2 gradients produces a linear gradient dynamics:

$$\begin{pmatrix} \mathbb{E}_{v_i}[v_i(\beta_i(v_i) - \widehat{\mathrm{br}}_i)] \\ \mathbb{E}_{v_i}[\beta_i(v_i) - \widehat{\mathrm{br}}_i] \end{pmatrix} = A(\theta_i^t - \theta_i^*),$$

where the matrix is defined as:

$$A = \frac{nw}{n-1} \cdot \begin{bmatrix} \frac{1}{3} & \frac{n-1}{2n} \\ \frac{1}{2} & \frac{n-1}{n} \end{bmatrix}.$$

This is a scaled version of the previous BR gradient's matrix, which has been proven to be positive-definite, and the condition number is in the order of $\Theta(1)$.

Therefore, the approximate BR gradient locally converges to BNE, at a speed of $\boxed{\mathcal{O}(\log \frac{1}{d})}$. $\qquad\qquad\square$

## D. Additional Experiments

To illustrate our method's generalizing ability to more general auction settings without known closed-form solutions, we consider asymmetric first-price auctions with $n > 2$ bidders, which generally lack closed-form solutions.

Here we reuse the setting of Figure 3 by replacing the second price with the first price, where bidders are equally divided into 2 types: the strong bidders with $U[0, 1]$ and the weak bidders with $U[0, 0.5]$. In the context of $n = 10$, we conducted experiments to plot the learned strategies of various methods across different random initializations in Figure 10.

As shown in the figures, the learned strategies of existing baselines (i.e., SM and Utility Grad) exhibit a classical slow-converging pattern: the strategies place positive bids $b_i > 0$ even as $v_i \to 0$. While the closed-form BNE solutions under this setting, to our best knowledge, have not been derived, we can still infer that this bidding behavior surely deviates from BNE, as better utility could be achieved by bidding zero when $v_i = 0$.

Conversely, strategies derived from our Approx BR method do not exhibit this issue. Furthermore, the learned strategy curves suggest that strong bidders with large values tend to bid more conservatively due to reduced competition, which aligns with the characteristics of the simplified 2-bidder setting's BNE solution (Kaplan & Zamir, 2012).

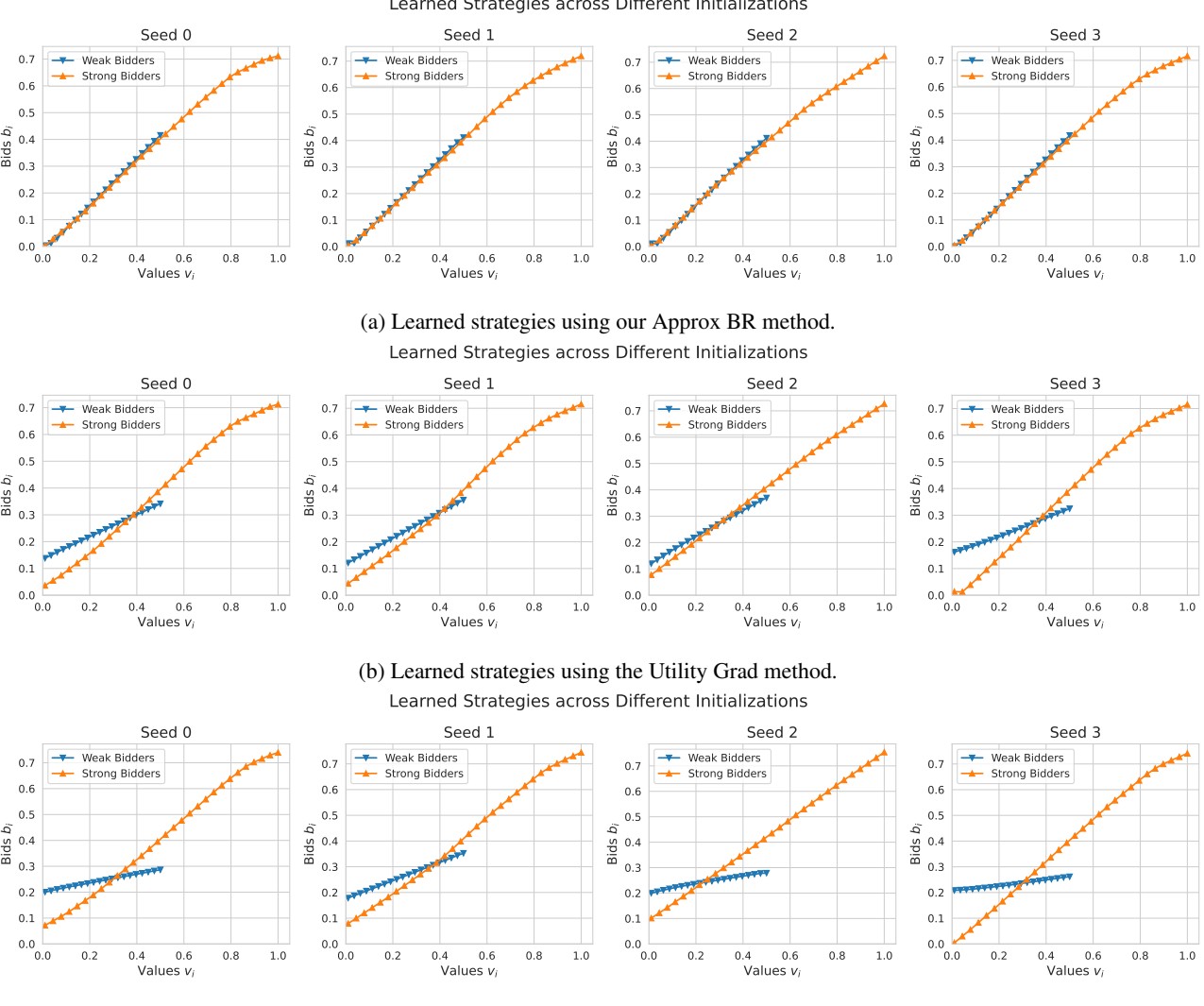

(a) Learned strategies using our Approx BR method.

(b) Learned strategies using the Utility Grad method.

(c) Learned strategies using the SM method.

*Figure 10.* Learned strategies of various methods across different random initialization under the asymmetric first price setting.

## E. Implementation Details

### E.1. Closed-Form BNE Solutions

Here we list the closed-form BNE solutions for the auction games considered in our experiments:

1. FP with symmetric uniform prior $U[0,1]$, risk-neutral bidders: $\beta_i^*(v_i) = \frac{n-1}{n} v_i$;

2. SP with symmetric uniform prior $U[0,1]$, risk-neutral bidders: $\beta_i^*(v_i) = v_i$;

3. FP with symmetric uniform prior $U[0,1]$, risk-averse bidders with utility defined in (19) and $\rho = 0.5$: $\beta_i^*(v_i) = \frac{2n-2}{2n-1} v_i$;

4. SP with symmetric uniform prior $U[0,1]$, risk-averse bidders: $\beta_i^*(v_i) = v_i$.

5. FP with asymmetric uniform prior $U[0,0.5]$ and $U[0,1]$, each type has 1 risk-neutral bidder: $\beta_1^*(v_1) = \frac{1}{3v_1}(1 - \sqrt{1 - 3v_1^2})$ for the weak bidder, $\beta_2^*(v_2) = \frac{1}{3v_2}(\sqrt{1 + 3v_2^2} - 1)$ for the strong bidder;

6. SP with asymmetric uniform prior $U[0,0.5]$ and $U[0,1]$, each type has $m$ risk-neutral bidders ($m \geq 2$): $\beta_i^*(v_i) = v_i$.

The solutions of (1), (2), (4), (5), and (6) can be directly found in (Krishna, 2010). As for the (3) case, we can directly apply the ODE (Equation (4.3) in (Krishna, 2010)):

$$\beta_i^{*\prime}(v_i) = \frac{u(v_i - \beta_i^*(v_i))}{u'(v_i - \beta_i^*(v_i))} \cdot \frac{g(v_i)}{G(v_i)}$$

where $u(x) = x^\rho$, $G(x) = \prod_{j \neq i} \mathcal{F}_j(x) = x^{n-1}$ and $g(x) = G'(x) = (n-1)x^{n-2}$. Thus the BNE strategy satisfies:

$$\beta_i^{*\prime}(v_i) = 2(v_i - \beta_i^*(v_i))\frac{n-1}{v_i},$$

with the boundary condition $\beta_i^*(0) = 0$, we have

$$\beta_i^*(v_i) = \frac{2n-2}{2n-1}v_i.$$

Additionally, we illustrate the learned strategies using our Approx BR method for the FP asymmetric setting with two value priors: $U[0, 0.5]$ and $U[0, 1]$ (*i.e.,* the No.5 setting above) in Fig. 11. The bidding strategies learned via our method closely align with the BNE solutions, confirming convergence even in this asymmetric setting.

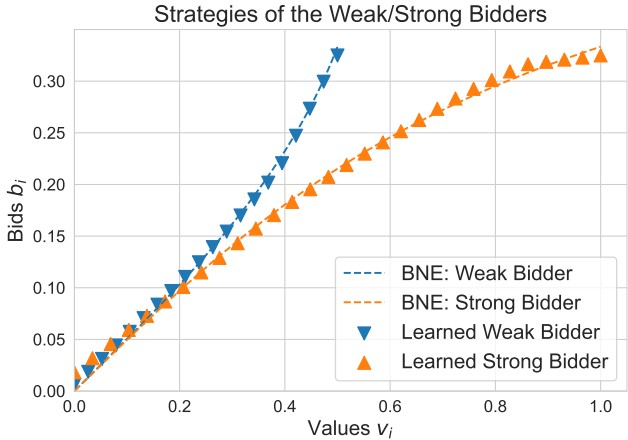

*Figure 11.* Theoretical BNE and the learned strategies with our Approx BR method in the asymmetric 2-player FP auction, which shows convergence in this asymmetric setting.

### E.2. Approximate BR Gradient Algorithm

The approximate BR gradient method is detailed in Algorithm 1.

### E.3. Codes & Hyper-Parameters

Since the official repository for the SM method (Kohring et al., 2023) does not provide instructions for reproducing their experiments, we have implemented the SM method independently. In the original SM paper, the *ex-post* utility is modified as:

$$u_i^{\mathrm{SM}}(v_i, b_i, b_{-i}) = (v_i - p_i^{\mathrm{SM}}) \cdot \frac{\exp(b_i/\lambda)}{\sum_{j=1}^n \exp(b_j/\lambda)}$$

where $p_i^{\mathrm{SM}} = \sum_{j \leq n} p_j$. In our implementation, we use $p_i^{\mathrm{SM}} = b_i$ for FP and $p_i^{\mathrm{SM}} = \max_{j \neq i} b_j$ for simplicity. Note that this modification doesn't change the smoothness and ability to approximate the original utility function of $u_i^{\mathrm{SM}}$. Our results align closely with the ones reported in the SM paper: The L2 error of our results is around 3e-3 and the SM paper's L2 is ranged in 4e-3~5e-3 when evaluated at the FP/SP symmetric uniform setting with 2 players. While improving the L2 result, the training time per iteration when $n = 2$ in our implementation is 0.012~0.013 seconds, which is comparable to the time range of 0.009~0.011 seconds reported in SM's paper.

---

**Algorithm 1** Stochastic Approximate BR Gradient Method

---

1: **Input:** Bidders $\{1 \ldots, n\}$, distributions $\mathcal{F}$, each bidder's strategy: $\beta_{\theta_i^{t=0}}$, learning rate $\alpha$, parameter $\gamma$, total steps $T$
2: **for** $t = 0$ **to** $T - 1$ **do**
3:     **for** $i = 1$ **to** $n$ **do**
4:         Sample $K$ values: $v_i^j \sim \mathcal{F}_i, j = 1, \ldots, K$ and corresponding bids $b_i^j = \beta_{\theta_i^t}(v_i^j)$
5:         Estimate the gradients of $\bar{u}_i^j$ via Eq. (11) and Eq. (15)
6:         **if** $\partial^2 \bar{u}_i^j / \partial b_i^2 < 0$ and $\partial \bar{u}_i^j / \partial b_i \approx 0$ **then**
7:             Calculate the approximate best response via second-order gradient: $\widehat{\mathrm{br}}_i^j = b_i^j - \left( \frac{\partial^2 \bar{u}_i^j}{\partial b_i^2} \right)^{-1} \frac{\partial \bar{u}_i^j}{\partial b_i}$
8:         **else**
9:             Calculate the approximate best response via first-order gradient: $\widehat{\mathrm{br}}_i^j = b_i^j + \gamma \frac{\partial \bar{u}_i^j}{\partial b_i}$
10:         **end if**
11:         Update strategy:

$$\theta_i^{t+1} \leftarrow \theta_i^t - \alpha \frac{1}{K} \sum_{j=1}^{K} \left( \beta_{\theta_i^t}(v_i^j) - \widehat{\mathrm{br}}_i^j \right) \cdot \nabla_{\theta_i} \beta_{\theta_i^t}(v_i^j)$$

12:     **end for**
13: **end for**
14: **Return:** Learned bidding strategies: $\beta_{\theta^{t=T}}$

---

For the estimation of pdf in the *ex-interim* utility's gradient (Eq. (11)), we utilize the kernel density estimation (KDE) approach with a Gaussian kernel:

$$\hat{f}_{m_i}(x) = \frac{1}{hN} \sum_{j=1}^{N} K\left( \frac{x - m_i^j}{h} \right),$$

where the $m_i^j, j = 1, \ldots, N$ are samples of market prices, and $K(\cdot)$ is the standard normal density function. The bandwidth $h$ in KDE is a hyperparameter that controls the approximation quality. For comparison, the SM gradient estimation uses a soft-max approximation for the winning indicator in the *ex-post* utility function:

$$\mathbb{I}(\text{bidder } i \text{ wins}) \approx \frac{\exp(b_i/\lambda)}{\sum_{j=1}^{n} \exp(b_j/\lambda)},$$

where the $\lambda$ is also a hyperparameter controlling the approximation quality. For the choices of these two parameters, we perform a grid search in the main experiment (Tab. 2) and use tuned values for the rest experiments.

As for other hyper-parameters during the training stages, we set the learning rate to 0.05 for all experiments except for the asymmetric SP auction, where we increased the learning rate to 0.2 for better results. Since all experiments considered in this work are under the independent private value (IPV) setting, this allows us to individually sample the $v_i$ and $v_{-i}$ values. For all experiments, we sample 256 values for $v_i$ and 10,000 value samples for $v_{-i}$. Since the value prior is mainly uniform, we use 10,000 equally divided value samples for L2 evaluation to avoid random noise.

Regarding the conditions for applying the second-order approximation in the approximate BR gradient method, we use the following criteria: 1. the 2nd-order approximation is not applied during the first 200 steps of training; 2. it's only enabled if $\partial^2 u_i / \partial b_i^2 \leq$ -1e-8 and $|\partial u_i / \partial b_i| \leq 0.01$ by default. These conditions ensure that the second-order approximation is used in the vicinity of a local optimum. The parameter $\gamma$ in first-order approximate best response is set to 1. Our code is open-sourced at: https://github.com/Hesse73/Approx-BR-Grad.

