# OpenReview forum: "Learning Bayesian Nash Equilibrium in Auction Games via Approximate Best Response"
_ICML.cc/2025/Conference — ICML 2025 poster_

### Official Review · Reviewer_RMDC · 2025-03-12

**Overall Recommendation:** 3

**Summary:**

This paper studies the problem of learning Bayesian Nash Equilibrium (BNE) in auction games as the number of bidders grows. The authors propose the Approximate Best Response Gradient method, including an analytic solution for gradient estimation to avoid the biased utility, and the Best Response Distance objective to address the slow convergence issue. A local convergence rate is proved to be independent of the number of bidders in symmetric auctions. Experiments across various auction formats, including different mechanisms, asymmetric value priors, and risk-averse utilities, show that the proposed method significantly accelerates convergence and enhances learning efficiency.

**Claims And Evidence:**

The claims made in the submission are supported by clear and convincing evidence.

**Essential References Not Discussed:**

None.

**Experimental Designs Or Analyses:**

The experimental designs and analysis sound.

**Methods And Evaluation Criteria:**

The proposed methods and evaluation criteria make sense for the problem.

**Other Comments Or Suggestions:**

NA

**Other Strengths And Weaknesses:**

NA

**Questions For Authors:**

NA

**Relation To Broader Scientific Literature:**

There have been several works proposing gradient-based approach to solve NE of some classes of games, and the proposed metric, Best Response Distance, is commonly used. Nevertheless, the gradient estimation is novel for auction games.

**Theoretical Claims:**

As far as I checked, the proofs are correct.

---

> ### Author Rebuttal · Authors · 2025-04-01
>
> > There have been several works proposing gradient-based approach to solve NE of some classes of games, and the proposed metric, Best Response Distance, is commonly used. Nevertheless, the gradient estimation is novel for auction games.
>
> Thank you for your positive feedback and for highlighting the innovative aspects of our work!
>
> Regarding the Best Response (BR) Distance metric, we would like to kindly point out that it differs from several existing metrics due to its focus on measuring **strategy-level** distance:
> $$
> \\|\beta_i(v_i) - \arg\max_b\bar u_i(v_i,b,\beta_{-i})\\|^2,
> $$
> where $\beta_i(\cdot)$ is the bidding strategy.
>
> This approach contrasts with other prevalent metrics, such as exploitability or the Nikado-Isoda function and its variants [1], which typically measure **utility-level** distance in a form like:
> $$
> \max_b \bar u_i(v_i,b,\beta_{-i}) - \bar u_i(v_i,\beta_i(v_i),\beta_{-i}),
> $$
> where $\bar u_i$ is the bidder's ex-interim utility.
>
> The strategy-level distance was chosen to mitigate slow local convergence issues associated with the **ill-conditioned utility function**, as detailed in Lemma 4.1.
> And we have proved that **BR-distance serves as an upper bound of the approximation factor $\epsilon$ in $\epsilon$-BNE** in Lemma 4.3, so optimizing it leads to a closer approximation to the BNE.
>
> However, if there are existing works that adopt a similar strategic approach akin to the BR-distance, we would be delighted to incorporate them into our references, as your suggestions are invaluable in ensuring comprehensive coverage of related research.
>
> Thank you again for recognizing the contributions of our work and for your valuable insights.
>
> ---
> **References**
>
> [1] Gemp, I., et. al. Approximating nash equilibria in normal-form games via stochastic op- timization. ICLR 2024

---

### Official Review · Reviewer_Be25 · 2025-03-12

**Overall Recommendation:** 3

**Summary:**

This paper investigates the problem of learning approximate ex-ante BNE in auction games under a publicly known prior distribution of bidder values. It proposes three new algorithms:

1. **Utility Grad**, which computes the gradient of bidders' utilities analytically using the CDF and PDF of the value distribution. However, as the authors point out, this method suffers from a low convergence rate that depends on the number of bidders.
2. **BR Gradient**, which optimizes the distance between bidders' current strategies and their best responses. Unlike Utility Grad, its convergence rate is independent of the number of bidders.
3. **Approximate BR Gradient**, which builds on BR Gradient by using a locally approximated best response.

Theoretical convergence rate analysis is conducted under the linear bidding assumption (Assumption 3.1). Empirically, the authors evaluate Utility Grad and Approximate BR Gradient when the bidding function is implemented using neural networks.

**Claims And Evidence:**

Yes

**Essential References Not Discussed:**

N/A

**Experimental Designs Or Analyses:**

In the experiment, the auction settings are restricted to cases with simple analytical solutions for BNE. More complex settings can be explored.

**Methods And Evaluation Criteria:**

Yes

**Other Comments Or Suggestions:**

See the discussions above.

**Other Strengths And Weaknesses:**

### Strengths:
1. The authors provide a detailed discussion of existing gradient methods for learning BNE in games.
2. None of the proposed methods introduce bias to the utility function.
3. The convergence rates of both BR Gradient and Approximate BR Gradient are independent of the number of bidders.

### Weaknesses:
1. The authors assume that the prior distribution is common knowledge, which is a stronger assumption than in previous works (e.g., Bichler et al., 2021), where only sampled data is available. A known prior distribution simplifies algorithm design.
2. While the CDF and PDF can be approximated from data, doing so requires additional samples. The paper does not discuss whether this approximation affects the convergence rate of the methods.
3. The theoretical results rely on Assumption 3.1, and the analysis of convergence to a locally approximate BNE is missing. This is particularly important for Approximate BR Gradient, as the algorithm is based on finding a locally approximate best response.

**Questions For Authors:**

N/A

**Relation To Broader Scientific Literature:**

The paper proposes a learning method to compute BNE of auction games, which is PPAD-hard.

**Theoretical Claims:**

The proofs are OK to me.

---

> ### Author Rebuttal · Authors · 2025-04-01
>
> ## More Settings with Unknown BNE
> We acknowledge that the experimental evaluation in our paper primarily focuses on auctions with known BNEs. This choice was made deliberately to allow for a clear and precise assessment of the learned strategies by comparing them against analytically derived solutions. Specifically, it enables us to quantify the error of the learned strategies $\beta_{\theta_i}$ using the $l2$-distances to the analytic solution $\beta_i^*$.
>
> To evalute our method in more complex settings without known BNEs, let's consider asymmetric first-price auctions with $n>2$ bidders, which generally lack closed-form solutions.
> We can reuse the setting of Figure 3 by eplacing the second price with first price, where bidders are equally divided into 2 types: the strong bidders with $U[0,1]$ and the weak bidders with $U[0,0.5]$.
>
> In the context of $n=10$, we conducted experiments to plot the learned strategies of various methods across different random initializations. The detailed **results are available in this [anonymous link](https://anonymous.4open.science/r/Figures-1718/plot_strategies.md)**.
>
> As shown in the figures, **the learned strategies of existing baselines (i.e., SM and UG) exhibit a classical slow-convergeing pattern**: the stratgies place positive bids $b_i > 0$ even as $v_i\to0$.
> While exact BNE solutions are unknown in these cases, we can infer that this bidding behavior **surely deviates from BNE**, as better utility could be achieved by bidding zero when $v_i=0$.
>
> Conversely, strategies derived from **our Approximate Best Response gradient method do not exhibit this issue**. Furthermore, the learned strategy curves suggest that strong bidders with large values tend to bid more conservatively due to reduced competition, which **aligns with the characteristics of 2-bidder setting's BNE solution [1]**.
>
> This result verifies the effectiveness in accelerating convergence of our method under more complex settings.
>
> ## Known Prior Assumption
> In fact, the referenced paper [2] also assumes the distribution to be common knowledge (Page 5, line 5: *"$f$ is assumed to be common knowledge"*).
> The assumption is basically a setting for auction game rules so that the bidders know how the values are sampled.
>
> In our method, as demonstrated in Algorithm 1 (Appendix D.2), the optimization process begins with each bidder independently sampling their own values, followed by gradient updates.
> During this procedure, **our algorithm does not explicitly access the distribution**.
> Instead, **it leverages sampled values to drive game optimization**, which is consistent with most existing methodologies.
>
> ## Sampling & Convergence Rates
> In this paper, we mainly focus on the **optimization convergence rates**, which means we are interested in how many update steps are required by different algorithms to achieve a similar target approximation level.
>
> As for the sampling complexity, the sampling process is a fundamental part of **both our gradient estimation, and and the existing ES or SM estimation methods**.
> Importantly, in our experiments, we ensured that all algorithms had access to **the same amount of data samples** for a fair comparison.
> The results, as presented in Table 2, indicate that **the wall-clock time for our gradient estimation is significantly less than that of SM-based estimations**.
> This efficiency gain is attributed to our analytic gradient computation, which alleviates the computational burden during backpropagation.
>
> Thus, while sampling is an intrinsic requirement for all compared methodologies, our approach still demonstrates superior efficiency in wall-clock time without compromising the quality of gradient estimations.
>
> We hope this clarification could addresses your concerns regarding the convergence rate and sampling impact.
>
>
> ## Convergence to Local BNE
> The Approximate BR (ABR) Gradient method **can indeed converge to the local approximate BNE**:
>
> As discussed in lines 300-310, the ABR method simplifies to the Utility Gradient (UG) method when employing the first-order approximation, with the second-order approximation only activated in the vicinity of the BNE.
> Therefore, the **ABR method shares the same convergence ability of the UG method when analyzing if it can converge to a local approximate BNE**.
>
> As established in **Proposition 3.2, the UG method can indeed converge to the BNE** under Assumption 3.1, substantiating the convergence capability of the ABR method as well.
>
> We will include this explanation in our revised version to ensure clarity.
> Thank you for emphasizing this issue!
>
> ---
> **Reference**
>
> [1] Kaplan, T. R. and Zamir, S. Asymmetric first-price auctions with uniform distributions: analytic solutions to the general case. Economic Theory, 50(2):269–302, 2012.
>
> [2] Bichler, M., et. al. Learning equilibria in symmetric auction games using artificial neural networks. Nature machine intelligence, 3(8):687–695, 2021.

---

### Official Review · Reviewer_7Vtz · 2025-03-14

**Overall Recommendation:** 3

**Summary:**

This paper presents the Approximate Best Response Gradient method for learning Bayesian Nash Equilibrium (BNE) in auction games. Auction plays a crucial role in many modern trading environments, including online advertising and public resource allocation, but computing BNE is computationally hard. Existing methods face challenges in gradient computation and optimization, especially in large-scale auctions.

This paper aims to address these challenges. First, they introduce an analytic solution for utility gradient estimation, avoiding the biased utility problem in existing methods. Second, they propose the Best Response Distance objective. By optimizing this objective, the proposed method achieves a local convergence rate of $O(\log (1/d))$, while the traditional method has a rate of $O(n^2\log (1/d))$. To reduce computational burdens, they further propose an approximate best response approach using local Taylor expansions.

Extensive experiments across various auction scenarios, including different mechanisms, asymmetric value priors, risk-averse utilities, and alternative gradient estimation approaches, demonstrate that the proposed method significantly accelerates convergence and enhances learning efficiency.

**Claims And Evidence:**

Yes.

**Essential References Not Discussed:**

No, to my knowledge.

**Experimental Designs Or Analyses:**

Yes. No issues have been found so far.

**Methods And Evaluation Criteria:**

Yes.

**Other Comments Or Suggestions:**

No.

**Other Strengths And Weaknesses:**

Strengths
1. The analytic gradient solution provides a more accurate way to estimate gradients compared to existing methods, which suffer from biased utility functions. This allows for more reliable learning of BNE strategies.
2. The new optimization objective leads to a significantly faster local convergence rate.
3. The paper provides a comprehensive theoretical analysis of the convergence rates of different methods.
4. The extensive experiments on different auction scenarios validate the superiority of the proposed method in terms of convergence speed and accuracy.

Weaknesses
1. The theoretical framework is mainly based on symmetric auctions with a uniform prior and a shared linear bidding strategy. It may not be directly applicable to more complex auction types and bidding strategies.
2. The closed-form solution for gradient estimation cannot generalize to other auction settings easily. Nowadays, most real-world online advertising scenarios do not use pure first-price or second-price auctions.
3. The lack of clear expression around Equation (10). The text fails to explicitly convey the concept of exploring the consequences of incorrect gradient calculation, which could potentially confuse readers. The sentence "the key issue..." is hard to parse.

**Questions For Authors:**

How to determine a good \gamma in Equation (16)?

**Relation To Broader Scientific Literature:**

This paper contributes to the BNE computation literature with a faster method. The result could be very useful for online advertising. However, since this paper focuses too much on first-price and second-price auctions only, it may lack of broader impact.

**Theoretical Claims:**

Yes. No issues have been found so far.

---

> ### Author Rebuttal · Authors · 2025-04-01
>
> ## Theoretical Assumptions
> Thanks for highlighting this point!
>
> First, we acknowledge that achieving **convergence in learning algorithms under general game settings is a challenging problem**, which is further compounded by unknown BNE solutions for such settings.
> **The focus of this work is on accelerating convergence**, so we opted to establish our theoretical results within the simplified framework, **consistent with previous research** [1,2].
>
> Despite the assumptions of asymmetric uniform prior and linear strategies for theoretical derivation, we've empirically verified the effectiveness of our method under **various auction scenarios with neural network strategies**, including different mechanisms, asymmetric value priors, risk-averse utilities, and alternative gradient estimation approaches, and demonstrated ***significantly accelerated convergence and enhanced learning efficiency*** as you've noted.
> Within our rebuttal experiments, we also extend our evaluation to settings with unknown BNEs (please refer to our response to Reviewer QBEq), which further validates the acceleration capabilities of our method in general cases.
>
> Moreover, there are **two key theoretical advancements that have broader applicability** beyond the mentioned assumptions:
> 1. **Gradient Estimation Technique**: We address the model bias issue in existing works through the analytic gradient based estimation.
> Notably, this method can be applied to general first-price (FP) and second-price (SP) auctions, **without relying on the mentioned uniform or symmetric assumptions**.
> 2. **Best Response (BR) Distance Objective**: This objective provides an upper bound to the approximation factor of $\epsilon$-BNE (Lemma 4.3), ensuring that optimizing this objective refines the BNE.
> Importantly, this result is **independent of uniform, symmetric, or specific FP/SP auction assumptions**, making it a viable objective for optimizing more general auction games. Furthermore, we have developed a practical approximation for the argmax operator in the BR-distance, proving its efficacy in the simplified auction setting, which could also inspire future research aimed at solving general auctions with similar approximations.
>
> We hope these clarifications could highlight the significance and potential impact of our contributions.
>
> ## Gradients & FP/SP
> First-price (FP) and second-price (SP) auctions are indeed prevalent in real-world applications. For instance, **Google Ads** employs **FP auctions** for their online advertisement services, having transitioned from **SP auctions** [3].
> We focus on these auctions due to their wide applications like exsting works [1,4].
>
> While it's true that real-world implementations of FP/SP auctions can include additional configurations, **such as reserve prices (bid floors), our gradient estimation approach is adaptable to such scenarios**.
>
> When introducing a reserve price $r$, the ex-iterim utility is modified as:
> - FP: $\bar u_i(v_i,b_i,\beta_{-i}) = \mathbb E_{v_{-i}}[(v_i - b_i)\cdot \mathbb I(b_i>\max\\{\beta_{-i}(v_{-i}),r\\})]$
> - SP: $\bar u_i(v_i,b_i,\beta_{-i}) = \mathbb E_{v_{-i}}[(v_i - \max\\{\beta_{-i}(v_{-i}),r\\})\cdot \mathbb I(b_i>\max\\{\beta_{-i}(v_{-i}),r\\})]$
>
> To estimate the gradients, we can **replace the original market price $m_{i} = \max_{j\neq i}b_j$ with a reserved version**: $m_{i}^r = \max\\{m_i, r\\} = \max\\{\beta_{-i}(v_{-i}),r\\}$.
> We can estimate the distribution of the reserved market price $m_i^r$ by sampling bids $b_{-i}$, and compute the pdf/cdf $f_{m_i^r}$ and $F_{m_i^r}$.
> **Then the gradient under reserve price can be estimated via Equation (11) by changing the distribution from $m_i$ to $m_i^r$**.
>
> This flexibility in gradient computation highlights its ability to generalize to more complex auction settings beyond pure FP/SP setups.
>
> ## Explaination on Eq. (10)
> Thanks for your valuable feedback, we will add explaination on Eq.(10) in our revised version:\
> In Eq. (10), the gradient is computed as $-\text{Pr(i wins)}\cdot \nabla_{\theta_i}\beta_{\theta_i}(v_i)$.
> Here $\text{Pr(i wins)}$ is the winning probability of bidder $i$, which remains positive unless $b_i = 0$.
> Since the gradient' coefficient $-\text{Pr(i wins)} < 0$ unless $b_i=0$, **the gradient for the bidder's bid $b_i=\beta_{\theta_i}(v_i)$ is consistently negative, unless $b_i=0$**.
> So the optimization procedure will continuouly reduce the bid $b_i$, **until reaching the stationary point $b_i=0$**.
> This is why the incorrect MC gradient estimation results in the zero-bidding problem.
>
> ## Hyperparameters
> We simply set $\gamma$ to 1 in our experiments (Line 1354).
>
> ## References
> [1] Convergence analysis of no-regret bidding algorithms in repeated auctions
>
> [2] On the convergence of learning algorithms in bayesian auction games
>
> [3] https://blog.google/products/admanager/update-first-price-auctions-google-ad-manager/
>
> [4] Enabling first-order gradient-based learning for equilibrium computation in markets

---

> > ### Comment · Reviewer_7Vtz · 2025-04-03
> >
> > Authors need to revise the paper as provided in the rebuttal.

---

> > > ### Author Response · Authors · 2025-04-03
> > >
> > > Thanks for your recognition of our work!
> > >
> > > We will ensure that the revised version includes the detailed explanations and the additional experiments in rebuttal.
> > > Thank you again for your time and insightful suggestions!

---

### Official Review · Reviewer_QBEq · 2025-03-14

**Overall Recommendation:** 3

**Summary:**

This paper introduces the Approximate Best Response Gradient method to efficiently learn Bayesian Nash Equilibrium (BNE) in auction games. It addresses the challenges of gradient computation and slow convergence in existing methods by using an analytic gradient solution and a novel Best Response Distance objective. The method achieves a local convergence rate independent of the number of bidders and demonstrates improved learning efficiency across several auction scenarios.

**Claims And Evidence:**

The non-convergence of best response dynamics is a well-known challenge for computing (Bayesian) Nash equilibrium in general games. Even when limited to auction games, the BNE of first price auction was unsolved for many decades. This work is making a valuable contribution in finding a response dynamics with certain convergence guarantees.

**Essential References Not Discussed:**

I don’t know any.

**Experimental Designs Or Analyses:**

See methods and evaluation criteria above.

**Methods And Evaluation Criteria:**

The evaluation of the proposed methods, however, seems a little bit limited. Only second price auctions and first price auctions are evaluated in the experiment section, while the BNEs are solved theoretically for both symmetric and asymmetric settings (except the risk-aversion case). It seems to me that the method should work for broader settings where the BNE is theoretically unknown. But the current experiment does not show any advantage of the proposed method in finding BNEs. Leaving me to doubt why this is an important problem.

**Other Comments Or Suggestions:**

N/A

**Other Strengths And Weaknesses:**

See Claims and Evidence above.

**Questions For Authors:**

No further questions.

**Relation To Broader Scientific Literature:**

It might be relevant to finding BNE for more general cases.

**Theoretical Claims:**

I didn’t verify all the details, but the theoretical part looks correct to me.

---

> ### Author Rebuttal · Authors · 2025-04-01
>
> ## General Settings with Unknown BNEs
>
> Thank you for your valuable feedback.
>
> Indeed, the experimental evaluation in our paper primarily focuses on auctions with known BNEs. This choice was made deliberately to **allow for a clear and precise assessment of the learned strategies** by comparing them against analytically derived solutions. Specifically, it enables us to quantify the error of the learned strategies $\beta_{\theta_i}$ using the $l2$-distances to the analytic solution $\beta_i^*$.
>
> However, we would like to emphasize that **this evaluation choice does not imply that our method cannot generalize to settings with unknown BNEs**.
> To illustrate, let's consider asymmetric first-price auctions with $n>2$ bidders, which generally lack closed-form solutions.
> We can reuse the setting of Figure 3 by eplacing the second price with first price, where bidders are equally divided into 2 types: the strong bidders with $U[0,1]$ and the weak bidders with $U[0,0.5]$.
>
> In the context of $n=10$, we conducted experiments to plot the learned strategies of various methods across different random initializations. **The detailed results are available in this [anonymous link](https://anonymous.4open.science/r/Figures-1718/plot_strategies.md)**.
>
> As shown in the figures, **the learned strategies of existing baselines (i.e., SM and UG) exhibit a classical slow-convergeing pattern**: the stratgies place positive bids $b_i > 0$ even as $v_i\to0$.
> While exact BNE solutions are unknown in these cases, we can infer that this bidding behavior **surely deviates from BNE**, as better utility could be achieved by bidding zero when $v_i=0$.
>
> Conversely, strategies derived from **our Approximate Best Response gradient method do not exhibit this issue**. Furthermore, the learned strategy curves suggest that strong bidders with large values tend to bid more conservatively due to reduced competition, which **aligns with the characteristics of the simplified 2-bidder setting's BNE solution [1]**.
>
> We hope these explanations and supplementary results alleviate your concerns regarding the significance of our work and illustrate its broader applicability.
>
> ---
> **Reference**
>
> [1] Kaplan, T. R. and Zamir, S. Asymmetric first-price auctions with uniform distributions: analytic solutions to the general case. Economic Theory, 50(2):269–302, 2012.

---

> > ### Comment · Reviewer_QBEq · 2025-04-07
> >
> > I might be missing something. Did you claim that the BNE of first price auction with $n = 10$ asymmetric bidders is unknown?

---

> > > ### Author Response · Authors · 2025-04-07
> > >
> > > Thank you for raising this clarification.
> > >
> > > The BNE in complex settings (e.g., first price auctions with $n > 2$ asymmetric bidders) can indeed be characterized by the corresponding differential equations (based on the first-order conditions).
> > > Our phrasing *"unknown BNEs"* was meant to highlight the **lack of closed-form solutions** in such cases, not to suggest that no theoretical characterization exists.
> > >
> > > We appreciate your feedback and apologize for any confusion caused by our wording.

---

### Decision · Program_Chairs · 2025-05-01

**Decision:**

Accept (poster)

**Comment:**

Overall this paper saw decent support from the reviewers (three weak rejects), I also think that the rebuttals clarified most of the reviewers' questions and concerns and augmented the papers contribution by presenting additional experiments. Given the interest in this topic in EC/ICML/NeurIPS recently, I think we can consider this paper for acceptance. If accepted the authors should address the reviewers' comments and incorporate the additional experiments from the discussion phase. There is also a very recent SODA 2025 paper by Ahunbay and Bichler, which the authors may want to discuss.